# Self-Supervised Evolution Operator Learning for High-Dimensional Dynamical Systems

**Giacomo Turri**[1]   **Luigi Bonati**[2]   **Kai Zhu**[2,3]   **Massimiliano Pontil**[1,4]   **Pietro Novelli**[1]

[1]Computational Statistics and Machine Learning, Italian Institute of Technology, Genoa, Italy
[2]Atomistic Simulations, Italian Institute of Technology, Genoa, Italy
[3]College of Pharmaceutical Sciences, Zhejiang University, Hangzhou, China
[4]AI Centre, University College London, London, United Kingdom

## Abstract

We introduce an end-to-end approach to learn the evolution operators of large-scale non-linear dynamical systems, such as those describing complex natural phenomena. Evolution operators are particularly well-suited for analyzing systems that exhibit spatio-temporal patterns and have become a key analytical tool across various scientific communities. As terabyte-scale weather datasets and simulation tools capable of running millions of molecular dynamics steps per day are becoming commodities, our approach provides an effective tool to make sense of them from a data-driven perspective. The core of it lies in a remarkable connection between self-supervised representation learning methods and the recently established learning theory of evolution operators. We deploy our approach across multiple scientific domains: explaining the folding dynamics of small proteins, the binding process of drug-like molecules in host sites, and autonomously finding patterns in climate data. Our code is available open-source at: `https://github.com/pietronvll/encoderops`.

## 1 Introduction

Dynamical systems are fundamental to understanding phenomena across a vast range of scientific disciplines, from physics and biology to climate science and engineering. Traditionally, scientists have modeled these systems by formulating differential equations from first principles. However, as systems grow in scale and complexity, this approach quickly becomes computationally burdensome and difficult to interpret (Anderson, 1972), hindering the study of large-scale phenomena. Simultaneously, advancements in data collection techniques and computational power have led to an explosion of available data from experiments (Hersbach et al., 2020; Chanussot et al., 2021) and high-fidelity simulations (Harvey et al., 2009; Abraham et al., 2015; Eastman et al., 2017; Bauer et al., 2015). This abundance of data makes data-driven approaches increasingly appealing for studying complex dynamics, with machine learning (Shalev-Shwartz & Ben-David, 2014) becoming a dominant paradigm for learning dynamical systems, largely focusing on predictive tasks such as forecasting. The recent revolution in data-driven weather modeling (Kurth et al., 2023; Bi et al., 2022; Lam et al., 2023; Kochkov et al., 2024) stands as a paradigmatic example of ML's power in handling complex spatio-temporal dynamics. Similarly, reinforcement learning (Sutton & Barto, 1998) has reimagined control theory by leveraging data-driven strategies to optimize system behavior. While these data-driven methods excel at prediction and simulation, there remains a significant gap in approaches that offer interpretability. In scientific contexts, merely predicting system behavior is often insufficient; understanding why a system evolves in a certain way is paramount. For instance, comprehending the dynamical shortcuts and bottlenecks happening through atomistic interaction is crucial for understanding why a drug binds to a specific target or fails to do so, a level of insight not typically provided by black-box predictive models.

A modeling paradigm particularly well-suited for interpretability is that of *evolution operators* (Lasota & Mackey, 1994; Applebaum, 2009). Under mild assumptions, dynamical systems and stochastic processes can be represented by a linear operator — a mathematical entity that maps functions to other functions. This operator-based approach offers multiple advantages. First, it linearizes the dynamics,

greatly simplifying tasks like forecasting and controller design. Second, these operators possess a spectral decomposition[1] (Reed & Simon, 1972), which expresses the system's complex dynamics as a linear combination of fundamental, coherent spatio-temporal modes (Molgedey & Schuster, 1994). Each mode represents a distinct, intrinsic pattern associated with a unique spatio-temporal structure defined in terms of growth or decay rates and oscillation frequencies. By identifying and analyzing these principal modes, researchers gain deep insights into the underlying mechanisms driving the system's macroscopic behavior, offering a structured, physically meaningful understanding.

Building on the understanding that evolution operators provide a powerful framework for interpretable analysis, significant effort has been directed towards learning these operators directly from data Kovachki et al. (2023). Data-driven approaches for this task emerged already in the early 2000s, including pioneering work utilizing transfer operators for analyzing stochastic processes in computational biophysics (Schütte et al., 2001), as well as the dynamic mode decomposition family of methods (Schmid, 2010) for deterministic systems via the Koopman operator. In the ensuing years, there has been a significant acceleration in machine learning methods for evolution operator learning, encompassing theoretical advances through kernel methods and powerful end-to-end deep learning approaches.

**Contributions.** In this work, we build upon these recent foundations, showing how evolution operator learning can be scaled to structured and high-dimensional dynamical systems. We formalize a principled end-to-end protocol that is amenable to GPU training and prove its equivalence to a self-supervised representation learning problem. Leveraging this link, we also show the transferability of our trained models in both molecular dynamics and climate settings. Code, data, and weights are made available open-source.

## 2 EVOLUTION OPERATORS AND HOW TO LEARN THEM

Evolution operator learning is a data-driven approach to characterizing dynamical systems, either stochastic, $x_{t+1} \sim p(\cdot|x_t)$, or deterministic, $x_{t+1} \sim \delta(\cdot - F(x_t))$. Throughout, we assume the dynamics to be Markovian, so that the evolution of $x_t$ depends on $x_t$ alone and not on the states at times $s < t$. If this assumption is not satisfied by $x_t$, a standard trick is to re-define the state as a context $c_t^H = f(x_t, x_{t-1}, \ldots, x_{t-H})$ with history length $H$, where $f$ can be a simple concatenation, or a learned sequence model (e.g., a recurrent neural network or transformer).

*Evolution operators* are defined as follows: for every function $f$ of the state of the system, $(\mathsf{E}f)(x_t)$ is the expected value of $f$ one step ahead in the future, given that at time $t$ the system was found in $x_t$

$$(\mathsf{E}f)(x_t) = \int p(dy|x_t)f(y) = \mathbb{E}_{y \sim X_{t+1}|X_t}[f(y)|x_t]. \tag{1}$$

Notice that $\mathsf{E}$ is an operator because it maps any function $f$ to another function, $x_t \mapsto (\mathsf{E}f)(x_t)$, and is *linear* because $\mathsf{E}(f + \alpha g) = \mathsf{E}f + \alpha \mathsf{E}g$. When the dynamics is deterministic, $\mathsf{E}$ is known as the *Koopman operator* (Koopman, 1931), while in the stochastic case it is known as the *transfer operator* (Applebaum, 2009).

Evolution operators fully characterize the dynamical system because knowing $\mathsf{E}$ allows us to reconstruct the dynamical law $p(\cdot|x_t)$. Indeed, for any subset of the state space $B \subseteq \mathcal{X}$, applying $\mathsf{E}$ to the indicator function of $B$, we have

$$(\mathsf{E}1_B)(x_t) = \int_B p(dy|x_t) = \mathbb{P}\left[X_{t+1} \in B|x_t\right].$$

An advantage of the operator approach over dealing directly with the conditional probability $p(\cdot|x_t)$ is that $\mathsf{E}$ acts linearly on the objects to which it is applied. This means that operators unlock an arsenal of tools from linear algebra and functional analysis, which would be unavailable otherwise. Arguably the most important of them is the spectral decomposition, allowing us to decompose $\mathsf{E}$, and hence the dynamics, into a linear superposition of dynamical modes. These ideas lie at the core of the celebrated Time-lagged Independent Component Analysis (Molgedey & Schuster, 1994; Pérez-Hernández et al., 2013), and Dynamical Mode Decomposition (Schmid, 2010; Kutz et al., 2016).

---

[1]A generalization of the eigenvalue decomposition of a matrix.

## 2.1 LEARNING E AND ITS SPECTRAL DECOMPOSITION FROM DATA

We now review the main approaches to learn the evolution operator and its spectral decomposition from a finite dataset of observations, with an emphasis on the least squares approach, which is essential to understand every other method as well.

A core idea of operator learning is that operators are defined by how they act *on a suitable linear space of functions*, similarly to how matrices are defined by their action on a basis of vectors. Of course, not every function $f$ is interesting, and this nicely parallels with the matrix example, where the most "interesting" directions are those that recover most of the variance in the data. Learning E, therefore, is usually cast as the following problem:

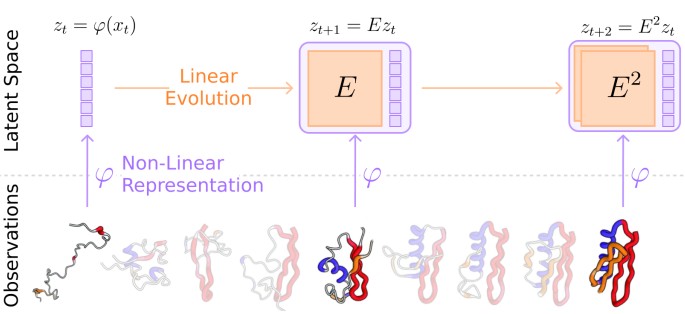

> Letting $\varphi(x) \in \mathbb{R}^d$ be a — learned or fixed — encoder of the state, find the best approximation of E *restricted* to the $d$-dimensional linear space of functions generated by $\varphi$, given the data.

In practice, the data is usually a collection of transitions $\mathcal{D} = (x_i, y_i)_{i=1}^N$, where it is intended that $x_i \sim \mathbb{P}[X_t]$ are sampled from a distribution of initial states, while $y_i \sim p(\cdot|x_i)$.

**Least squares.** In this approach the encoder $\varphi$ is a frozen, that is non-learnable, dictionary of functions, and we are interested in approximating the action of E on functions of the form $f(x) = \langle w, \varphi(x) \rangle$ for every $w \in \mathbb{R}^d$. To this end, one minimizes the empirical error between the true conditional expectation $\mathbb{E}_{y \sim X_{t+1}|X_t}[\langle w, \varphi(y) \rangle | x]$, and a linear model $\langle Ew, \varphi(x) \rangle$, where the matrix $E \in \mathbb{R}^{d \times d}$ identifies the restriction of the evolution operator to the linear span of the dictionary:

$$\frac{1}{N} \sum_{i=1}^N (\langle w, \varphi(y_i) \rangle - \langle Ew, \varphi(x_i) \rangle)^2 \leq \frac{1}{N} \sum_{i=1}^N \|\varphi(y_i) - E^\top \varphi(x_i)\|^2 + \lambda \|E\|^2. \quad (2)$$

On the right-hand side, we assumed $\|w\| \leq 1$, used the Cauchy–Schwarz inequality, and added a ridge penalty. The minimizer of (2) can be computed in closed form (Korda & Mezić, 2018; Kostic et al., 2022, and references therein) as

$$E_\varphi = (C_X + \lambda \mathsf{Id})^{-1} C_{XY}, \quad \text{with } C_{XY} = \frac{1}{N} \sum_{i=1}^N \varphi(x_i) \varphi(y_i)^\top \text{ and } C_X = C_{XX}. \quad (3)$$

In the limit of infinite data, $N \to \infty$, and infinitely dimensional encoders, $d \to \infty$, the least squares estimator converges (Korda & Mezić, 2018) in the strong operator topology to the evolution operator E, and similar (but weaker) asymptotic convergence results are proved for its spectrum.

**Mode decomposition.** The spectral decomposition of E is approximated by expressing the least-squares estimator in its eigenvectors' basis $E_\varphi = Q\Lambda Q^{-1}$, where the columns of $Q = [q_1, \cdots, q_d]$ are the eigenvectors of $E_\varphi$, and $\Lambda$ is a diagonal matrix of eigenvalues. In this basis, the expected value in the future for a function $f(x) = \langle w, \varphi(x) \rangle$ is expressed as

$$\mathbb{E}_{y \sim X_{t+1}|X_t}[f(y)|x] \approx \langle E_\varphi w, \varphi(x) \rangle = \langle Q\Lambda Q^{-1} w, \varphi(x) \rangle = \sum_{i=1}^d \lambda_i \langle q_i, \varphi(x) \rangle (Q^{-1} w)_i. \quad (4)$$

The spectral decomposition expresses the transition $x_t \to x_{t+1}$ as a sum of *modes* of the form $\lambda_i \langle q_i, \varphi(x) \rangle (Q^{-1} w)_i$, each of which can be broken down into three components:

1. The eigenvalues $\lambda_i$ determine the time scales of the transition. Indeed, applying the evolution operator $s$ times to analyze the transition $x_t \to x_{t+s}$ leaves (4) unchanged, except that each $\lambda_i$ becomes $\lambda_i^s$. Writing $\lambda_i^s = \rho_i^s e^{is\omega_i}$ in polar coordinates, reveals that the modes decay exponentially over time with rate $\rho_i$, while oscillating at frequency $\omega_i$.

2. The initial state $x$ influences the decomposition through the factor $\Psi_i(x) = \langle q_i, \varphi(x) \rangle$. This coefficient captures how strongly the state $x$ aligns with the $i$-th mode. When $q_i$ corresponds to an eigenvalue with slow decay, i.e., $|\lambda_i| \approx 1$, the term $\Psi_i(x)$ serves as a natural quantity for clustering states into *coherent* or *metastable* sets.

3. The coefficient $(Q^{-1}w)_i$, in turn, indicates how the function represented by the vector $w$ relates to the $i$-th mode. This connection makes it possible to link the dynamical patterns to specific functions — or *observables* – thereby deepening our understanding of the system.

**Kernel methods.** Leveraging the kernel trick, one can learn evolution operators by deriving a closed-form solution of (2) in terms of kernel matrices whose elements are of the form $k(x_i, x_j) = \langle \varphi(x_i), \varphi(x_j) \rangle$, with $k(\cdot, \cdot)$ a suitable kernel function. Thanks to the theory of reproducing kernel Hilbert spaces, this class of methods is backed up by statistical learning guarantees, such as the ones derived in (Kostic et al., 2022; 2023; Nüske et al., 2022). Similarly to the least-squares approach, one also approximates the spectral decomposition of E via kernel methods, and this task captured quite a lot of attention from researchers in this area, see (Williams et al., 2015; Kawahara, 2016; Klus et al., 2020; Das & Giannakis, 2020; Alexander & Giannakis, 2020; Meanti et al., 2023).

**Deep learning.** In contrast to the previous approaches, where the encoder $\varphi$ is prescribed, a number of methods proposed to approximate E from data with end-to-end schemes including $\varphi$ as a learnable neural network. Since learning E ultimately entails learning its action on the linear space spanned by $\varphi$, it is appealing to choose an encoder capturing the most salient features of the dynamics. To this end, one can train $\varphi$ via an *encoder-decoder* scheme as proposed in (Takeishi et al., 2017; Lusch et al., 2018; Otto & Rowley, 2019; Azencot et al., 2020; Wehmeyer & Noé, 2018; Frion et al., 2024) or with *encoder-only* approaches as in (Li et al., 2017; Mardt et al., 2018; Yeung et al., 2019; Kostic et al., 2024b; Federici et al., 2024; Jeong et al., 2025).

In encoder-decoder schemes, $\varphi$ is trained alongside a decoder network, minimizing a combination of prediction and reconstruction errors[2]. Yet, while minimizing a reconstruction loss biases the model towards accurate forecasts of the near future, recent work on model-based reinforcement learning, where one is instead interested in long-term behaviours, suggests that the presence of a decoder is detrimental (Lyu et al., 2023; Schwarzer et al., 2020; Hansen et al., 2024) to control tasks. Similarly, (Balestriero & LeCun, 2024) showed how features learned by reconstruction are both uninformative for perception and hardly transferable.

On the other hand, the competitive advantage of evolution operators over techniques such as (Kurth et al., 2023; Lam et al., 2023; Pfaff et al., 2021; Sanchez-Gonzalez et al., 2020; Li et al., 2020) lies in their spectral decomposition, useful for interpretability, reduced order modeling, and control tasks. Encoder-only approaches follow this intuition and prioritize approximating the spectral decomposition of E over the raw forecasting performances. Concretely, this is accomplished via loss functions that are minimized when $\varphi$ spans the leading singular space of E. Clearly, once an encoder has been trained, it can be transferred to similar dynamical systems, as we demonstrate in 4.2 and 4.3.

In this work, we propose an encoder-only method based on a loss function originally designed for self-supervised representation learning. Our approach is numerically stable, scales efficiently, enables transfer across related systems, and can incorporate structural priors, e.g., graph-based encoders, beyond the reach of classical DMD approaches. Though our approach is broadly applicable, we mainly focus on applications involving interpretability and model reduction of scientific dynamical systems, highlighting how ML evolution operators can help in advancing fundamental science. Recent works in RL (Lyu et al., 2023; Schwarzer et al., 2020; Rozwood et al., 2023; Novelli et al., 2024) suggest that our approach can be relevant for control tasks, but we leave this for future work.

## 3 LEARNING EVOLUTION OPERATORS VIA SELF-SUPERVISION

As discussed above, we are interested in the evolution operator

$$(\mathsf{E}f)(x_t) = \mathbb{E}_{y \sim X_{t+1}|X_t}[f(y)|x_t] = \mathbb{E}_{y \sim X_{t+1}}\left[\frac{p(y|x_t)}{p(y)}f(y)\right], \tag{5}$$

---

[2]Notice that trying to minimize the prediction error (2) alone immediately leads to a *representation collapse* with $\varphi$ mapping every input to 0 to obtain a prediction error of 0.

where in the last equality we expressed the expectation in the form of an importance sampling estimator with respect to the probability of the future state $\mathbb{P}(X_{t+1})$. In so doing, we link the evolution operator $\mathsf{E}$ to the density ratio

$$r(x_t, x_{t+1}) = \frac{p(x_{t+1}|x_t)}{p(x_{t+1})}. \tag{6}$$

The core of our operator-learning scheme, Alg. 1, is to optimize a model for the density ratio (6) parametrized as the bilinear form $\langle \varphi(x_t), P\varphi(x_{t+1}) \rangle$, similar to van den Oord et al. (2019). Here, $\varphi$ is a $d$-dimensional encoder, while $P$ is a linear *predictor* layer which, as discussed below, equals the action of $\mathsf{E}$ on the linear subspace of functions spanned by $\varphi$, up to a known linear transformation.

We minimize the $L^2$ error between the density ratio and our bilinear model $\langle \varphi(x_t), P\varphi(x_{t+1}) \rangle$:

$$
\begin{aligned}
\varepsilon(\varphi, P) &= \mathbb{E}_{(x,y)\sim X_t \otimes X_{t+1}} \left[ (r(x,y) - \langle \varphi(x), P\varphi(y) \rangle)^2 \right] \\
&= \mathbb{E}_{(x,y)\sim X_t \otimes X_{t+1}} \left[ \langle \varphi(x), P\varphi(y) \rangle^2 \right] - 2\mathbb{E}_{(x,y)\sim (X_t, X_{t+1})} \left[ \langle \varphi(x), P\varphi(y) \rangle \right] + \text{cst.}, \tag{7}
\end{aligned}
$$

where $\mathbb{E}_{X_t \otimes X_{t+1}}$ is the expected value between the product of the marginals $X_t$ and $X_{t+1}$[3].Estimating the squared term in (7) via U-statistics (Hoeffding, 1992) and foregoing the constant term, we finally get to the empirical loss

$$\hat{\varepsilon}(\varphi, P) = \frac{1}{N(N-1)} \sum_{i \neq j} \langle \varphi(x_i), P\varphi(y_j) \rangle^2 - \frac{2}{N} \sum_{i=1}^{N} \langle \varphi(x_i), P\varphi(y_i) \rangle. \tag{8}$$

The loss function (8) was originally proposed for self-supervised contrastive learning in HaoChen et al. (2021; 2022). Indeed, noticing that $\langle \varphi(x), P\varphi(y) \rangle$ can be interpreted as a measure of similarity between $x$ and $y$, the first term of (8) minimizes the similarity between randomly chosen $i \neq j$ (i.e., *negative*) pairs, while the second term maximizes the similarity of consecutive (i.e., *positive*) pairs. The loss (8) has also been applied in reinforcement learning (Ren et al., 2023), causal estimation (Sun et al., 2025), and recently Lu et al. (2024) showed that it belongs to a wide class of contrastive learning losses defined by Csiszár $f$-divergences. Concurrently,Wang et al. (2022); Ryu et al. (2024); Kostic et al. (2024a); Jeong et al. (2025) applied it to approximate the SVD of linear operators.

### 3.1 THEORETICAL PROPERTIES OF OUR APPROACH.

Our model $\langle \varphi(x_t), P\varphi(x_{t+1}) \rangle$ is characterized by the presence of a linear predictor $P$, and by a shared encoder between $x_t$ and $x_{t+1}$, in contrast to the more agnostic choice $\langle \varphi(x_t), \psi(x_{t+1}) \rangle$ adopted by HaoChen et al. (2021); Wang et al. (2022); Ryu et al. (2024); Kostic et al. (2024a); Jeong et al. (2025). Our choice is deliberate, and we now prove a number of theoretical results highlighting how our parametrization is particularly apt for evolution operator learning, with the predictor layer $P$ having a key role. Every lemma in this section is proved in Appendix A. The first observation was already noticed in Wang et al. (2022), and provides a direct link between (7) and the evolution operator regression formalism developed in Kostic et al. (2022).

**Lemma 1.** *Let $\varphi : \mathcal{X} \to \mathbb{R}^d$ be an encoder whose components are square-integrable with respect to both $\mu$ and $\nu$, and let $\mathsf{E}$ be a Hilbert-Schmidt evolution operator. Then, the loss function (7) is equivalent to the following operator learning loss:*

$$\varepsilon(\varphi, P) = \| \mathsf{E} - \sum_{i,j} \varphi_i \otimes P_{ij}\varphi_j \|_{\mathsf{HS}}^2.$$

We highlight that when the operator $\mathsf{E}$ is not Hilbert-Schmidt, the loss function (8) can still be linked to operator learning. In this more general scenario, the loss promotes encoders $\varphi$ displaying both a strong dynamical response $\mathsf{E}\varphi$, and a good approximation of the true dynamics, see Appendix A.1.

Plugging our model back into (5), the evolution operator gets parametrized as $(\mathsf{E}f)(x_t) \approx \mathbb{E}_{y\sim X_{t+1}} [\langle \varphi(x_t), P\varphi(y) \rangle f(y)]$, and if the function $f$ is in the linear span generated by the encoder $f(y) = \langle \varphi(y), w \rangle$, we can simplify the expression above as

$$(\mathsf{E}f)(x_t) = \langle \varphi(x_t), P \left( \mathbb{E}_{y\sim X_{t+1}} [\varphi(y)\varphi(y)^\top] \right) w \rangle = \langle \varphi(x), PC_Y w \rangle,$$

---

[3]That is, the product measure $\mathbb{P}[X_t] \otimes \mathbb{P}[X_{t+1}]$

where we introduced the covariance of the futures $C_Y = \mathbb{E}_{y \sim X_{t+1}}[\varphi(y)\varphi(y)^\top]$. Thus, the linear predictor $P$ parametrizes the approximation of the evolution operator $\mathsf{E}$ over the finite-dimensional space generated by the state representation $\varphi$. We remark that to be sure that a prescribed function $f$ lies in the span of the encoder, one can add it as a non-trainable component of the architecture $\varphi(x) = [\mathrm{NN}(x), f(x)]$, as done in Appendix B.1. Alternatively, one can compute its least-squares approximation $\hat{f}_\varphi$ on the features spanned by $\varphi$, and use that in place of $f$. The following Lemma shows that when the predictor $P$ is optimal, one recovers the least squares estimator (3).

**Lemma 2.** *For any fixed $\varphi$, the predictor $P$ minimizing (7) can be computed in closed form $P_* = C_X^{-1} C_{XY} C_Y^{-1}$, and the model for the evolution operator is given by*

$$E_\varphi = P_* C_Y = C_X^{-1} C_{XY} = Eq. \text{ (3) with } \lambda \to 0, \tag{9}$$

*coinciding with the least-squares estimator (3).*

The final interesting fact about (7) is its relation to the VAMP score (Wu & Noé, 2020), originally introduced for representation learning of molecular kinetics. In particular, the VAMP-2 score can be defined in terms of covariances as

$$\mathrm{VAMP}_2(\varphi) = \|C_X^{-1/2} C_{XY} C_Y^{-1/2}\|_{\mathsf{HS}}^2. \tag{10}$$

**Lemma 3.** *For any fixed $\varphi$, let $P_*$ the optimal predictor of $\varepsilon(\varphi, P)$, as in Lemma 2. Then, the following holds true:*

$$\varepsilon(\varphi, P_*) = -\|C_X^{-1/2} C_{XY} C_Y^{-1/2}\|_{\mathsf{HS}}^2 = -VAMP_2(\varphi).$$

Our loss function, therefore, matches the negative VAMP-2 score when $P$ is optimal. Compared to methods that directly maximize the VAMP score, such as (Mardt et al., 2018), however, our approach does not require matrix inversions in the computation of the loss (Wu & Noé, 2020), an operation which is unwieldy and prone to instabilities[4] in large-scale applications. Instead, the loss function (8) is written in terms of simple matrix multiplications, making it perfect for GPU-based training.

### 3.2 PRACTICAL IMPLEMENTATION

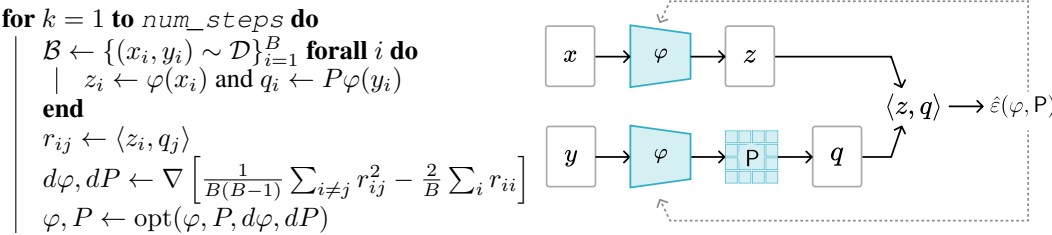

**for** $k = 1$ **to** `num_steps` **do**
    $\mathcal{B} \leftarrow \{(x_i, y_i) \sim \mathcal{D}\}_{i=1}^B$ **forall** $i$ **do**
      $z_i \leftarrow \varphi(x_i)$ and $q_i \leftarrow P\varphi(y_i)$
    **end**
    $r_{ij} \leftarrow \langle z_i, q_j \rangle$
    $d\varphi, dP \leftarrow \nabla \left[ \frac{1}{B(B-1)} \sum_{i \neq j} r_{ij}^2 - \frac{2}{B} \sum_i r_{ii} \right]$
    $\varphi, P \leftarrow \mathrm{opt}(\varphi, P, d\varphi, dP)$
**end**

**Algorithm 1:** A pair of consecutive observations $(x, y)$ from a dynamical system are mapped to representations $z$ and $q$ via an embedding function $\varphi$. The representation $q$ is also processed by a predictor $P$. The algorithm iteratively optimizes $\varphi$ and $P$ using the contrastive objective (8) based on the similarity $\langle z, q \rangle$.

The implementation of our method, summarized in Alg. 1, follows standard self-supervised learning procedures (Chen et al., 2020; Grill et al., 2020; Zbontar et al., 2021; Chen & He, 2021). There, a *positive pair* of data-points, in our case a pair of consecutive observations of the dynamical system, are processed through an encoder network $\varphi$, and optionally a predictor network, which in our case is a simple linear layer $P$. We apply simplicial normalization (Lavoie et al., 2022) to the outputs of the embedding $\varphi$. To keep our implementation as close to the theoretical insights as possible, we didn't concatenate additional projection heads to the encoder $\varphi$, as suggested in (Chen et al., 2020; Grill et al., 2020; Zbontar et al., 2021; Chen & He, 2021). Furthermore, because of the identity (9) we kept $P$ linear, but it is worth mentioning that tiny MLPs might be employed as predictors instead.

---

[4]Backpropagation through inversions may lead to gradient explosion (Golub & Pereyra, 1973).

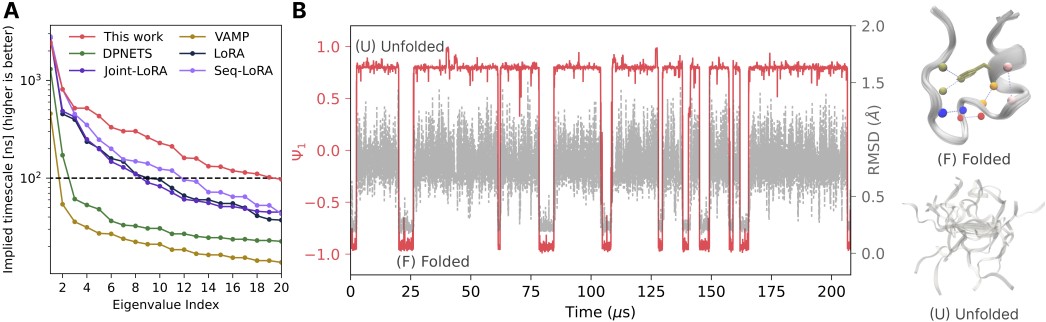

Figure 1: *Trp-Cage folding*. **A**: Implied timescales derived from different baseline methods. Higher implied timescales are associated with more accurate approximations of the slow-modes. **B**: Time series of the leading eigenfunction $\Psi_1$ (red, left axis) alongside RMSD (gray, right axis), capturing transitions between folded and unfolded states. Representative snapshots of each state are shown. In the folded structure, key hydrogen bonds identified as relevant by the LASSO model are highlighted.

Once a representation $\varphi$ is learned, we model the evolution operator E via the least-squares estimator $E_\varphi$ from (3). To compute it, one can make use of the closed form expression (3) by computing the covariances at the end of the training of $\varphi$, as done in Kostic et al. (2024b); Jeong et al. (2025). This two-step procedure, however, requires a full forward pass over the full training dataset, which is impractical for large problems. Another option is to use (9), but this again requires the evaluation of the covariance, and might be suboptimal whenever $P$ isn't yet converged to the true minimizer $P_*$. In our implementation, instead, we kept two buffers for $C_X$ and $C_{XY}$, which are updated online during the training loop via an exponentially moving average of the batch covariances. At the end of the training, we use buffers to compute $E_\varphi$ as in (3). In Appendix B.5, we show that covariances updated online during training converge to an accurate approximation of the true covariances, and yield identical (or slightly better) results compared to re-evaluating the covariances from scratch.

## 4 EXPERIMENTS

We now put to the test our method on high-dimensional dynamical systems from both molecular dynamics and climate domains. Our focus is on assessing the capability of the method to decompose complex dynamics and to evaluate the generalizability of the learned representations. In Appendix B.1 we also report an additional experiment on the Lorenz '63 system (Lorenz, 1963), illustrating how the method can also be used for small forecasting tasks.

### 4.1 HIGH-RESOLUTION DYNAMICAL MODELING OF PROTEIN FOLDING

The Trp-Cage miniprotein is a widely studied benchmark for protein folding due to its small size and fast dynamics (Lindorff-Larsen et al., 2011). Previous works, including SRV-based Markov State Models (Sidky et al., 2019) and GraphVAMPNet (Ghorbani et al., 2022), have modeled Trp-Cage dynamics using coarse-grained representations, where the state of the system is defined by the small subset of 20 $C_\alpha$ atoms in the backbone of the protein. Our approach allows us to scale to a more expressive molecular representation based on all 144 heavy atoms, employing the SchNet (Schütt et al., 2017) graph network architecture as the encoder $\varphi$. After training, we calculate the eigenvalue decomposition of the evolution operator as described in Sec. 2. As shown in Fig. 1B, the leading eigenfunction $\Psi_1(x) = \langle q_1, \varphi(x) \rangle$ correlates strongly with the system's root-mean-square deviation (RMSD) from the folded structure, confirming that $\Psi_1$ encodes the folding-to-unfolding transition. Clustering the molecular configurations according to the values of $\Psi_1$ reveals a clear separation between folded and unfolded ensembles (see snapshots in Fig. 1B).

To interpret the nature of this slow mode, we regress $\Psi_1$ against a library of physically meaningful descriptors—specifically, hydrogen bond interactions across residue pairs—using a sparse LASSO model (Brunton et al., 2016; Zhang et al., 2024; Novelli et al., 2022). This analysis reveals a network of hydrogen bonds stabilizing the folded state, including contributions from side-chain interactions

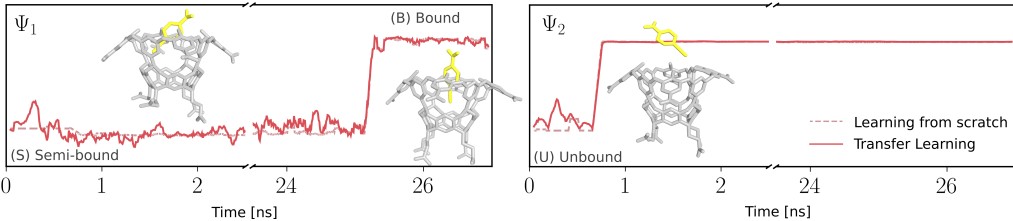

Figure 2: *Calixarene binding.* Eigenfunctions $\Psi_1$ (left) and $\Psi_2$ (right) capture ligand transitions from unbound (U) to semi-bound (S) and bound (B) states. The model using a representation transferred from other ligands (solid line) closely matches one trained from scratch (dashed).

that would be invisible to coarse-grained dynamical models such as (Ghorbani et al., 2022). Finally, we note that the implied timescales[5] $\tau_i$ derived from the leading eigenvalues of the learned operators are influenced by both the choice of representation and learning loss. The LoRA baselines from Jeong et al. (2025) use a similar loss function, but do not share the encoder $\varphi$ between $x_t$ and $x_{t+1}$. VAMP, from Mardt et al. (2018) and DPNETS from Kostic et al. (2024b), on the other hand, minimize different loss functions. According to the variational principle for Markov processes (Wu & Noé, 2020; Noé & Nüske, 2013), higher implied timescales indicate a better approximation of the system's true slow dynamics (see Fig. 1A).

### 4.2 LEARNING TRANSFERABLE REPRESENTATIONS FOR THE BINDING OF SMALL MOLECULES

Our second case study focuses on the binding of small molecules to a calixarene-based system (Yin et al., 2017), which is often used as a simplified model to study the dynamical processes relevant, for instance, in drug design. Our baseline is obtained by using Alg. 1 to train an encoder $\varphi$ on molecular dynamics data describing the binding dynamics of a single molecule (G2) to the host system. As in the previous example, we employ a SchNet architecture for $\varphi$. As shown in Fig. 2, the slowest dynamical mode, captured by the dominant eigenfunction $\Psi_1$, is associated with a transition between a semi-bound configuration and the fully bound state. Structural inspection reveals that this intermediate state corresponds to a misaligned pose of the guest, caused by the presence of a water molecule occupying the binding pocket. The second eigenfunction $\Psi_2$ instead resolves the unbound-to-bound transition. Our findings align with previous works (Rizzi et al., 2021), where water occupancy was identified as a key kinetic bottleneck in host–guest interactions.

We now turn to a key question: can a representation trained on one set of molecular systems generalize to others? This capability is essential for scalable modeling in applications like drug discovery, where retraining a model for every new compound is prohibitive. To test the transferability of the representations $\varphi$ trained with our method, we trained the encoder on molecular dynamics simulations for two molecules (G1 and G3), and used it to analyze the binding dynamics of a *different ligand* (G2). Using the frozen encoder, we compute the evolution operator of (G2) via (3), and examine its dominant eigenfunctions. Remarkably, the transferred representation successfully recovers the key dynamical modes of the binding process of (G2) without having seen it during the representation learning phase. In particular, it recovers both the entry of the guest molecule into the host cavity and its final locked configuration ( Fig. 2). This result illustrates that our self-supervised model learns features that are not only informative but also transferable across molecular systems.

### 4.3 PATTERNS IN GLOBAL CLIMATE

Finally, we test our method on climate data. Specifically, we aim to retrieve El Niño–Southern Oscillation (ENSO), one of the most influential sources of interannual climate variability (Diaz & Markgraf, 2000; Callahan & Mankin, 2023), arising from coupled ocean–atmosphere dynamics in the tropical Pacific (Bjerknes, 1969; Philander, 1983). Characterizing ENSO remains a central goal in climate science, particularly in the context of its potential changes under global warming

---

[5]The implied timescale can be computed from the eigenvalues as $\tau_i = -\Delta t / \log(\lambda_i)$, where $\Delta t$ is the time lag between two consecutive observations.

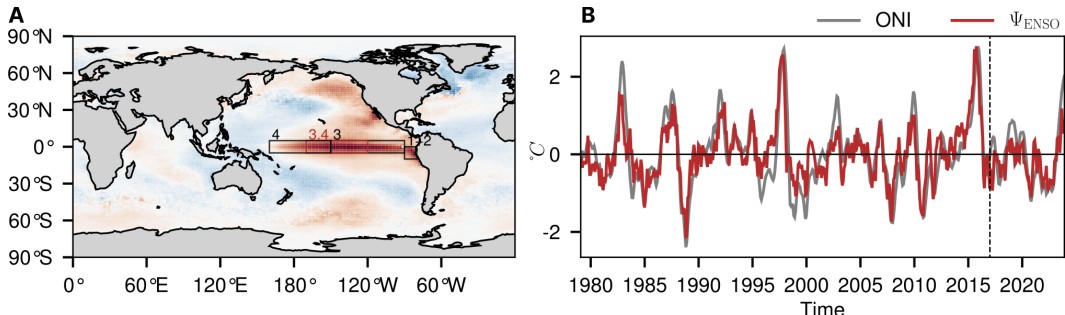

Figure 3: ENSO mode retrieved with our method. **A**: Mode associated with the second leading eigenfunction, highlighting dominant activation in the tropical Pacific. Boxes indicate standard ENSO monitoring zones. **B**: Right eigenfunction corresponding to the second leading eigenvalue, compared to the ONI index. The vertical line marks the split between training and validation sets.

(McPhaden et al., 2006; Cai et al., 2021).ENSO is conventionally characterized by monthly-averaged sea surface temperature (SST) anomalies, denoted as SST*, computed following the procedure described in (NCP Center). The SST fields are obtained from the ORAS5 reanalysis (Zuo et al., 2019) and provided through the ChaosBench dataset (Nathaniel et al., 2024). However, the dataset comprises only 540 snapshots, which may hinder effective model training. To overcome this, analogous to the drug design experiment described in Sec. 4.2, we adopt a transfer learning strategy using a longer synthetic trajectory generated by the Community Earth System Model (CESM) (Hurrell et al., 2013), consisting of 12,598 samples. The objectives of this experiment are twofold: (i) to determine whether our method can retrieve ENSO dynamics, and (ii) to assess whether representations $\varphi$ learned from simulated data can be effectively transferred to real-world climate observations.

A convolutional neural network-based encoder is trained using the simulated SST* fields, after which the learned representation $\varphi$ is applied to real data. The transfer operator is then estimated following (3), using the period 1979–2016 for training and 2017–2023 for validation, and subsequently subjected to spectral decomposition to extract the dominant modes. As expected, this procedure recovers modes corresponding to known climate periodicities, such as annual oscillations (see Tab. 2). Remarkably, one of the leading nontrivial modes (second in magnitude) clearly reflects ENSO dynamics. The associated right eigenfunction exhibits a strong Pearson correlation ($r = 0.82, p < .001$) with the Oceanic Niño Index (ONI) (Fig. 3B), a widely used metric for ENSO monitoring (Glantz & Ramirez, 2020), while the associated spatial mode shows dominant activation over the tropical Pacific (Fig. 3A). Importantly, our method generalizes effectively to unseen data, successfully detecting the 2023 El Niño event within the validation set. It is worth noting that training the same model directly on observational data also recovers the ENSO mode; however, the correlation between the associated right eigenfunction and ONI is weaker ($r = 0.71, p < .001$). Additionally, we compared our method against VAMPNets (Mardt et al., 2018) and DPNets (Kostic et al., 2024b), with results indicating that our approach achieves stronger correlation in capturing the ENSO mode (see Appendix B.4).

This experiment underscores the model's ability to autonomously identify complex climate phenomena in an unsupervised manner without prior localization (unlike previous approaches (Froyland et al., 2021; Lapo et al., 2025)). Importantly, the transfer learning approach enables the model to leverage knowledge from large, high-quality simulations to mitigate for the scarcity of observational data, thereby enabling a more robust extraction of complex patterns such as ENSO. These findings highlight the ability of our approach to learn a robust and generalizable representation, effectively transferring knowledge from synthetic simulations to real-world observations

## 5 CONCLUSION

In this work, we proposed an end-to-end framework for learning evolution operators and their spectral decomposition. Our method scales effectively to large and complex systems, making it a practical tool for uncovering physically meaningful patterns in their dynamics. By leveraging a connection between contrastive learning objectives and the spectral properties of evolution operators, we break new ground on the transfer of dynamical representations. Our experiments on atomistic and climate

systems demonstrate the versatility of our approach and its generalization capabilities. Looking ahead, this connection opens the door to more expressive learning architectures, robust training strategies, and broader applications in scientific discovery and control.

**Limitations.** Due to the nature of our experiments, evaluation was more qualitative than typical in ML; benchmarks specifically targeting the accuracy of the spectral decomposition are, to the best of our knowledge, not yet available.

## REPRODUCIBILITY STATEMENT

**Theory.** Our theoretical claims are supported by complete proofs provided in Appendix A.
**Code.** All code used in this study is available at `https://github.com/pietronvll/encoderops`. Detailed experimental procedures and implementation are described in Appendix B.
**Datasets.** Instructions for generating, downloading, or requesting the datasets are included in Appendix B.

## ACKNOWLEDGMENTS

This work was partially funded by the European Union - NextGenerationEU initiative and the Italian National Recovery and Resilience Plan (PNRR) from the Ministry of University and Research (MUR), under Project PE0000013 CUP J53C22003010006 "Future Artificial Intelligence Research (FAIR)", and European Project ELIAS N. 101120237. We acknowledge ISCRA for awarding this project access to the LEONARDO supercomputer, owned by the EuroHPC Joint Undertaking, hosted by CINECA (Italy).

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

## A  PROOFS OF THE THEORETICAL CLAIMS.

Define $\nu = \mathbb{P}[X_t]$, the distribution of the initial states in our dataset, and $\mu = \mathbb{E}_{x \sim \nu}[p(\cdot|x)] = \mathbb{P}[X_{t+1}]$ the distribution of the evolved states. In practice, $\nu$ can be the following:

- If a simulator is available, $\nu$ can be *any* distribution of initial states, and $\mu$ is obtained by a single step of the simulator on data from $\nu$.
- If one samples trajectories of length $T$ from an initial distribution $\mathbb{P}[X_1]$, then $\nu = \frac{1}{T} \sum_{i=1}^{T-1} \mathbb{P}[X_i]$.
- If — as in molecular dynamics, or the Lorenz 63 example below — one samples from an *invariant* distribution $\pi$ such that $\mathbb{P}[X_t] = \pi \implies \mathbb{P}[X_{t+1}] = \pi$, one has $\nu = \mu = \pi$.

The evolution operator $\mathsf{E}$ maps functions from $L^2(\mu)$ into $L^2(\nu)$, that is $\mathsf{E} : L^2(\mu) \to L^2(\nu)$. Notice that since we allow for general initial and evolved distributions, respectively $\nu$ and $\mu$, our method does *not* require $\mathsf{E}$ to be associated to a stationary, nor ergodic dynamical system, as often the case in the theoretical literature, see e.g. (Mezić, 2005, Section 2.3) or Kostic et al. (2022). We will now show this simple equivalence:

**Lemma 1.** *Let $\varphi : \mathcal{X} \to \mathbb{R}^d$ be an encoder whose components are square-integrable with respect to both $\mu$ and $\nu$, and let $\mathsf{E}$ be a Hilbert-Schmidt evolution operator. Then, the loss function (7) is equivalent to the following operator learning loss:*

$$\varepsilon(\varphi, P) = \|\mathsf{E} - \sum_{i,j} \varphi_i \otimes P_{ij}\varphi_j\|_{\mathsf{HS}}^2.$$

*Proof.* The Lemma was already proved in Wang et al. (2022, Lemma 4.1), or Kostic et al. (2024a, Theorem 1). Here we provide a self-contained proof. Since the encoder is square-integrable in both $\mu$ and $\nu$ — $\varphi_i \in L^2(\mu)$ and $L^2(\nu)$ — we can define the linear operators

$$\Phi_\mu : L^2(\mu) \to \mathbb{R}^d \qquad f \mapsto f = (\langle f, \varphi_i \rangle_{L^2(\mu)})_{i=1}^d.$$

$$\Phi_\nu^* : \mathbb{R}^d \to L^2(\nu) \qquad z \mapsto \sum_{i=1}^d \varphi_i(\cdot)z_i.$$

By direct substitution of the definition above, it follows that

$$\|\mathsf{E} - \Phi_\nu^* P \Phi_\mu\|_{\mathsf{HS}}^2 = \|\mathsf{E} - \sum_{i,j} \varphi_i \otimes P_{ij}\varphi_j\|_{\mathsf{HS}}^2.$$

---

[6]Version 1 of Jul 9, 2025.

Now, let's notice that by direct calculation one obtains

$$\Phi_\nu \Phi_\nu^* = \mathbb{E}_\nu \left[ \varphi(x)\varphi(x)^\top \right] \qquad \Phi_\nu \mathsf{E} \Phi_\mu^* = \mathbb{E}_\rho \left[ \varphi(x)\varphi(y)^\top \right],$$

where $\rho(dx, dy) = p(dy|x)\nu(dx)$ is the joint distribution of $(X_t, X_{t+1})$.

By the definition of the Hilbert-Schmidt norm, we have

$$
\begin{aligned}
\|\mathsf{E} - \Phi^* P \Phi\|_{\mathsf{HS}}^2 &= \|\mathsf{E}\|_{\mathsf{HS}}^2 - 2\mathrm{Tr}\left[\mathsf{E}^* \Phi_\nu^* P \Phi_\mu\right] + \mathrm{Tr}\left[\Phi_\mu^* P^\top \Phi_\nu \Phi_\nu^* P \Phi_\mu\right] \\
&= \|\mathsf{E}\|_{\mathsf{HS}}^2 - 2\mathrm{Tr}\left[\Phi_\mu \mathsf{E}^* \Phi_\nu^* P\right] + \mathrm{Tr}\left[\Phi_\mu \Phi_\mu^* P^\top \Phi_\nu \Phi_\nu^* P\right] \\
&= \|\mathsf{E}\|_{\mathsf{HS}}^2 - 2\mathbb{E}_{(x,y)\sim\rho}\left[\mathrm{Tr}\left[\varphi(y)\varphi(x)^\top P\right]\right] + \mathbb{E}_{(x,y)\sim\mu\otimes\nu}\left[\mathrm{Tr}\left[\varphi(y)\varphi(y)^\top P^\top \varphi(x)\varphi(x)^\top P\right]\right] \\
&= \|\mathsf{E}\|_{\mathsf{HS}}^2 - 2\mathbb{E}_{(x,y)\sim\rho}\left[\langle\varphi(x), P\varphi(y)\rangle\right] + \mathbb{E}_{(x,y)\sim\mu\otimes\nu}\left[\langle\varphi(x), P\varphi(y)\rangle^2\right],
\end{aligned}
$$

where we repeatedly used the cyclic property of the trace. $\qquad\square$

The following Lemma shows that when $P$ is optimal with respect to (7), then it recovers the least squares estimator (3).

**Lemma 2.** *For any fixed $\varphi$, the predictor $P$ minimizing* (7) *can be computed in closed form $P_* = C_X^{-1} C_{XY} C_Y^{-1}$, and the model for the evolution operator is given by*

$$E_\varphi = P_* C_Y = C_X^{-1} C_{XY} = Eq. \text{ (3) with } \lambda \to 0, \tag{9}$$

*coinciding with the least-squares estimator* (3).

*Proof.* The proof follows by noticing that $\varepsilon(\varphi, P)$ is convex in $P$. Taking the gradient (see, for example (Minka, 2000)) one has:

$$
\begin{aligned}
\nabla_P \varepsilon(\varphi, P) &= -2\mathbb{E}_{(x,y)\sim\rho}\left[\varphi(y)\varphi(x)^\top\right] + 2\mathbb{E}_{(x,y)\sim\mu\otimes\nu}\left[\varphi(y)\varphi(y)^\top P^\top \varphi(x)\varphi(x)^\top\right] \\
&= -2C_{YX} + 2C_Y P^\top C_X
\end{aligned}
$$

As the problem is convex, the global minimum $P_*$ is attained when $\nabla_P \varepsilon(\varphi, P_*) = 0$. This condition is equivalent to solve the equation

$$-2C_{YX} + 2C_Y P_*^\top C_X = 0.$$

By multiplying the expression above by $C_X^{-1}$ on the right and $C_Y^{-1}$ on the left, re-arranging it, and taking the transpose of everything, we finally get

$$P_* = C_X^{-1} C_{XY} C_Y^{-1}.$$

$\qquad\square$

The following Lemma shows the equivalence of (7) and the VAMP-2 loss of Wu & Noé (2020); Mardt et al. (2018)

**Lemma 3.** *For any fixed $\varphi$, let $P_*$ the optimal predictor of $\varepsilon(\varphi, P)$, as in Lemma 2. Then, the following holds true:*

$$\varepsilon(\varphi, P_*) = -\|C_X^{-1/2} C_{XY} C_Y^{-1/2}\|_{\mathsf{HS}}^2 = -VAMP_2(\varphi).$$

*Proof.* By noticing that the loss function (7) can be equivalently rewritten as

$$\varepsilon(\varphi, P) = \mathrm{Tr}[P^\top C_X P C_Y - 2P C_{YX}],$$

and substituting the optimal predictor $P_* = C_X^{-1} C_{XY} C_Y^{-1}$ from Lemma 2 inside this expression, we immediately obtain the identity. $\qquad\square$

A.1   ON THE HILBERT-SCHMIDT ASSUMPTION, AND BEYOND

In the main text, we assumed the evolution operator E to be Hilbert-Schmidt, which immediately guarantees the well-posedness of the proposed loss function (7). Lemma 1, indeed, implies that for any Hilbert–Schmidt E it holds

$$\varepsilon(\varphi, P) = \|E - \sum_{i,j} \varphi_i \otimes P_{ij}\varphi_j\|_{\mathsf{HS}}^2 < \infty.$$

The Hilbert-Schmidt assumption is valid in a broad class of stochastic systems, particularly when the transition kernel exhibits smoothing properties. In atomistic simulations, for instance, the presence of a finite temperature results in a Gaussian smoothing that makes E Hilbert-Schmidt. As an illustrative example, consider the overdamped Langevin dynamics

$$X_{t+1} = X_t - \nabla V(X_t)\,\Delta t + \mathcal{N}(0, \sigma)\sqrt{\Delta t},$$

where $V$ is a potential function, and $\mathcal{N}(0, \sigma)$ denotes an isotropic Gaussian with mean 0 and variance $\sigma$ proportional to the system's temperature. Assuming that the data is sampled from the equilibrium distribution $\pi(x)dx \propto e^{-\beta V(x)}dx$, we compute

$$\|E\|_{\mathsf{HS}}^2 = \sum_i \|Ee_i\|_2^2 = \sum_i \int \Big| \int p(y \mid x)\, e_i(y)\, \pi(dy) \Big|^2 \pi(dx)$$

$$= \int p(y \mid x)^2\, \pi(y)\pi(x)\, dy\, dx \propto \int \Big| \exp\Big(-\frac{\|y - \nabla V(x)\|}{2\sigma^2}\Big)\Big|^2 \pi(y)\pi(x)\, dy\, dx < \infty$$

where $e_i$ are elements of an orthonormal basis of $L^2(\pi)$, and the third equality follows from Parseval's identity.

The Hilbert-Schmidt assumption, however, is violated in important deterministic dynamical systems, such as those governed by fluid dynamics equations Mezić (2005). Remarkably, the empirical loss (8) still admits a precise operator-theoretic interpretation when E is merely a bounded operator. Indeed, let $P_\varphi : L^2(\mu) \to L^2(\mu)$ denote the orthogonal projector onto the subspace spanned by the encoder $\varphi$. By definition, $P_\varphi = \Phi_\mu^\dagger \Phi_\mu$, where $\Phi_\mu$ is as in Lemma 1. With a slight abuse of the notation in (7), we define the abstract loss

$$\varepsilon(\varphi, P) = -2\,\mathrm{Tr}[E^*\Phi_\nu^* P\Phi_\mu] + \|\sum_{i,j} \varphi_i \otimes P_{ij}\varphi_j\|_{\mathsf{HS}}^2,$$

whose empirical estimator exactly coincides with (8), the loss which we *actually* optimized in our experiments. Now, since $\varphi$ spans a finite-dimensional subspace, both $P_\varphi$ and $EP_\varphi$ are finite rank, hence Hilbert–Schmidt. In particular, $EP_\varphi$ is the restriction of E to the subspace generated by $\varphi$. An immediate calculation shows that

$$\varepsilon(\varphi, P) = \|EP_\varphi - \sum_{i,j} \varphi_i \otimes P_{ij}\varphi_j\|_{\mathsf{HS}}^2 - \|EP_\varphi\|_{\mathsf{HS}}^2, \tag{11}$$

where besides basic algebraic manipulations the result is obtained using the ciclicity of the trace, and the relation $\Phi_\mu P_\varphi = \Phi_\mu(\Phi_\mu^\dagger \Phi_\mu) = \Phi_\mu$.

The first term on the right-hand side of (11) is familiar, and represents the error incurred by the model $\sum_{i,j} \varphi_i \otimes P_{ij}\varphi_j$ in approximating the restriction of E to the subspace spanned by $\varphi$. This error can be linked to the least-squares approach discussed in Sec. 2, see, for example Korda & Mezić (2018, Theorem 1). Minimizing it with respect to $\varphi$ leads to representation collapse, since $\varphi(x) = 0$ for all $x$ trivially minimizes it. Our learning objective (11), instead, avoids collapse through the second term, $-\|EP_\varphi\|_{\mathsf{HS}}^2$, which can be interpreted as follows. Without loss of generality, write $P_\varphi = \sum_{i=1}^d e_i \otimes e_i$, where $e_i$ form an orthonormal basis of $\mathrm{span}(\varphi_1, \ldots, \varphi_d)$. By definition of Hilbert-Schmidt norm one has

$$\|EP_\varphi\|_{\mathsf{HS}}^2 = \sum_{i=1}^d \|Ee_i\|_2^2 = \sum_{i=1}^d \mathbb{E}_{x\sim\nu}\big[(Ee_i)(x)^2\big].$$

Now notice that interpreting $e_i$ as a probe we have at our disposal to observe the system[7], the term $(Ee_i)(x)$ quantifies the "dynamical response" read by our probe, given that the system was prepared

---

[7]Functions of the state of the systems are commonly referred to as *observables*, too.

Table 1: Forecasting errors and training times for the Lorenz '63 example (20 independent runs). Note that for LinLS and KRR is reported the total fitting time while for the other methods the epoch time is reported. Best results are highlighted in bold.

| | Ours | LinLS | KRR | VAMPNets | DPNets | DAE | CAE |
|---|---|---|---|---|---|---|---|
| RMSE ($\times 10^{-2}$) | **0.49**±**0.24** | 1.29±0.00 | 2.10±0.00 | 0.78±0.12 | 0.58±0.11 | 0.77±0.12 | 2.58±0.19 |
| MAE ($\times 10^{-2}$) | **0.32**±**0.24** | 0.84±0.00 | 1.27±0.00 | 0.46±0.08 | 0.36±0.08 | 0.55±0.08 | 1.95±0.14 |
| Time (ms) | 181.1±40.1 | **.4**±**.1** | $(25.3±0.2)10^3$ | 165.5±10.8 | 190.7±41.5 | 166.8±9.10 | 408.5±41.9 |

to be in state $x$. The quantity $\|\mathsf{EP}_\varphi\|_{\mathsf{HS}}^2$, therefore, measures the average strength of such responses, implying that the second term in the loss promotes encoders $\varphi$ whose span captures observables with the highest possible dynamical variability. To close the discussion, we highlight that for Hilbert-Schmidt operators, the observables with the highest dynamical response are precisely the leading singular functions, and the loss function (7) is indeed minimized when $\varphi$ spans the leading singular space of the evolution operator $\mathsf{E}$, see (Kostic et al., 2024a, Theorem 1).

## B  EXPERIMENTAL DETAILS

The experiments have been performed on the following hardware:

- 1 Node with 32 cores Ice Lake at 2.60 GHz, 4 × NVIDIA Ampere A100 GPUs, 64 GB and 512 GB RAM.

- 1 Node with 20 cores Xeon Silver 4210 at 2.20 GHz, 4 × NVIDIA Tesla V100 GPUs, 16 GB and 384 GB RAM.

- A workstation equipped with a i7-5930K CPU at 3.50 GHz, 2 × NVIDIA GeForce GTX TITAN X GPUs, 12 GB and 32 GB of RAM.

### B.1  ADDITIONAL EXPERIMENT: LORENZ '63

We evaluated our method on the Lorenz '63 system (Lorenz, 1963), a classical example of a chaotic dynamical system governed by three coupled ordinary differential equations. To validate the performance of our approach, we tested it on a one-step-ahead forecasting task, and we analyzed the learned dynamical modes. Because of the low-dimensionality of the state $x_t$, we appended it as a non-learnable feature of the encoder $\varphi(x_t) = [\mathrm{MLP}(x_t), x_t]$ to ensure that the forecasting target–the state itself–lies in the linear space of functions spanned by $\varphi$ by design. The learnable part of the encoder consisted of a small multi-layer perceptron (MLP).

In Tab. 1, we compare the performance of the estimator $E_\varphi$ from (3), with an encoder $\varphi$ trained according to Alg. 1, against several baseline models. These include Linear Least Squares (LinLS), Kernel Ridge Regression (Kostic et al., 2022) (KRR) with a Gaussian kernel, VAMPNets (Mardt et al., 2018), DPNets (Kostic et al., 2024b), Dynamic Autoencoder (Lusch et al., 2018) (DAE), and Consistent Autoencoder (Azencot et al., 2020) (CAE). To ensure a fair comparison, we matched the encoder architecture for VAMPNets, DPNets, DAE and CAE, while decoders of DAE and CAE were defined as MLPs symmetric to their respective encoders. For KRR, the rank was set equal to the latent dimensionality used in the deep learning models.

The results on the forecasting task demonstrate that, although our model is not specifically designed for prediction, it achieves the best performance among all considered methods. Finally, we verified that the leading eigenfunctions obtained by our approach correctly identify coherent sets on the stable attractor (see Fig. 4).

**Training details.** We generated a single long trajectory of 15,000 time steps using the `kooplearn` 1.1.3 implementation of Lorenz '63 dynamical system, with default parameters. To ensure convergence to a system's attractor (Tucker, 1999), we discarded the first 1,000 time steps. Also, to obtain approximately time-independent segments for training, validation and testing, we further discarded 1,000 time steps between each split. In total, 10,000 time steps were used for training, and 1,000 time steps each for validation and testing.

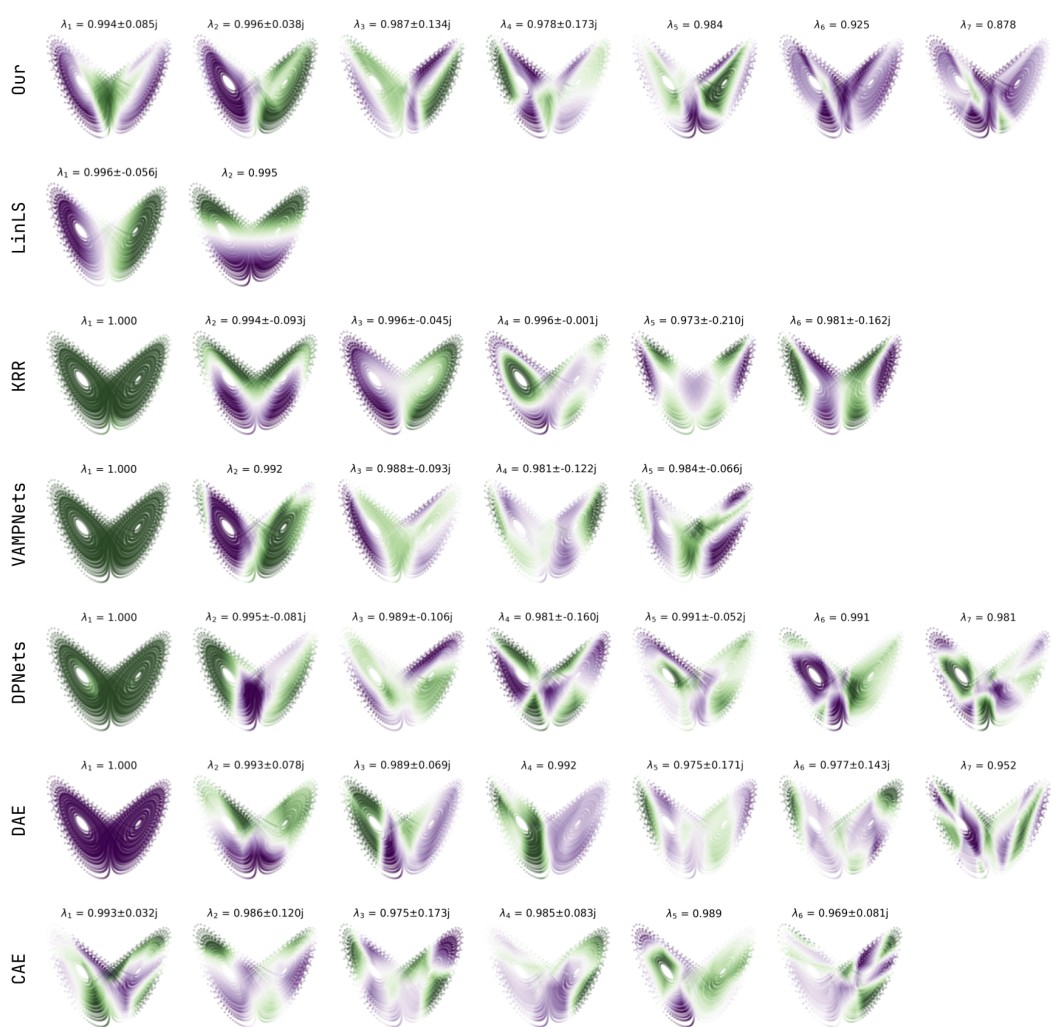

Figure 4: Leading eigenfunctions computed by our and baseline approaches. Each row corresponds to a different method, and each column shows an eigenfunction ordered by decreasing eigenvalue magnitude.

Our encoder consisted of an MLP with an input layer of size 3, two hidden layers with 16 units each, and an 8-dimensional latent space, using ReLU activation functions. The model was trained for 100 epochs using the AdamW optimizer, with an initial learning rate of $10^{-3}$ decayed to $10^{-4}$ via a cosine schedule, a batch size of 512, and a lag time of 10 time steps.

**Baseline methods.** We compared our approach against the following baseline models:

- **Linear Least Squares (LinLS).** A linear regression model trained directly on the raw input features without any nonlinear transformation or latent representation.

- **Kernel Ridge Regression (KRR) (Kostic et al., 2022).** We trained a KRR model with a Gaussian kernel, using the bandwidth estimated via the median heuristic (Garreau et al., 2017). The model was trained with a rank of 8, a Tikhonov regularization parameter of $10^{-6}$, and using Arnoldi iterations.

- **VAMPNets (Mardt et al., 2018).** Trained using the same MLP encoder as ours, with the VAMP-2 loss loss and centered covariances.

- **DPNets (Kostic et al., 2024b).** Trained using the same MLP encoder as ours, with the relaxed DP loss and centered covariances.

- **Dynamic Autoencoder (DAE) (Lusch et al., 2018).** Trained with the same MLP encoder architecture as in our approach; the decoder was defined symmetrically. The loss components for reconstruction, prediction, and linear evolution were equally weighted (all set to 1).

- **Consistent Autoencoder (Azencot et al., 2020).** Trained with the same MLP encoder architecture as in our approach; the decoder was defined symmetrically. The CAE loss weights for reconstruction, prediction, backward prediction, linear evolution, and consistency were all set to 1.

For all deep learning-based baselines (VAMPNets, DPNets, DAE, and CAE), models were trained for 100 epochs using a batch size of 512, and a lag time of 10 time steps. VAMPNets and DPNets used the AdamW optimizer with a learning rate of $10^{-4}$ and $10^{-2}$, respectively; DAE and CAE used the Adam optimizer with a learning rate of $10^{-3}$. All baselines were implemented using `kooplearn` 1.1.3.

**Additional analysis.** In Fig. 4, we show the leading eigenfunctions of the transfer operators computed using our method and the baseline approaches. These visualizations highlight qualitative differences in the learned spectral structures, offering insight into the dynamics captured by each method. The leading eigenfunction of KRR, VAMPNets, DPNets, and DAE is constant and associated with the stable attractor. Our method, LinLS, and KRR, find an eigenfunction with eigenvalue $\approx .996$ which clearly separates the two lobes of the attractor.

## B.2 PROTEIN FOLDING

**Training details.** We used data from (Lindorff-Larsen et al., 2011), which can be requested directly to De Shaw Research and are available without charge for academic usage. Our encoder consisted of a SchNet (Schütt et al., 2017) graph neural network with 3 interaction blocks, 16 RBF functions and an hidden dimension of 64. The model was trained with an AdamW optimizer with starting learning rate of $10^{-2}$ decaying to $10^{-4}$ with a cosine schedule, using the `mlcolvar` (Bonati et al., 2023) library.

**Additional analysis.** To understand to what mode is associated the leading eigenfunction $\Psi_1$, in Fig. 5 we correlated it with two physical quantities associated with the folding, which are the Root-Mean-Square-Deviation (RMSD) and the Radius of Gyration, see Fig. 5. Furthermore, to obtain a finer understanding, we used sparse linear models to approximate the CVs via LASSO regression. This yields a surrogate model which is a linear combination of a few physical descriptors, hence interpretable. To choose the regularization strength, we computed the Mean Square Error of the surrogate model versus the number of features, see Fig. 6.

We performed LASSO regression on a set of contact functions determining the presence of hydrogen bonds. The features selected by this procedure, as well as a snapshot of the protein where these features are highlighted, are reported in Fig. 7. Interestingly, some of the selected features pertain to side-chain interactions, a piece of information that would have been impossible to get using only $C_\alpha$ atoms to train the encoder.

## B.3 LIGAND BINDING

**Simulations details.** We selected a subset of host-guest systems for the SAMPL5 challenge (Bannan et al., 2016; Yin et al., 2017) to evaluate our method's performance, including three ligands (G1, G2, G3) and the octa-acid calixarane host (OAMe). Simulations were run in GROMACS 2024.5 (Abraham et al., 2015) patched with PLUMED 2.9.3 (Tribello et al., 2014). Systems were built using the GAFF (Wang et al., 2004) force field with RESP (Bayly et al., 1993) charges, solvated in a cubic TIP3P (Jorgensen et al., 1983) water box $40.27\,\text{Å}$ length, containing 2100 water molecules. System charge balanced with $Na^+$ ions. Our timestep is $2\,\text{fs}$ and the temperature is set to $300\,\text{K}$ via a velocity rescale thermostat (Bussi et al., 2007) with a coupling time of $0.1\,\text{ps}$. All simulations aligned the host's vertical axis $h$ with the box axis and centered coordinates on virtual atom V1. All production simulations were initiated from the dissociated state of each ligand. Trajectories were terminated when the ligand fully rebounded into the binding pocket (defined as host-guest distance $h < 6\,\text{Å}$). For each ligand, we performed 10 independent production trajectories, with coordinates saved every 500 steps.

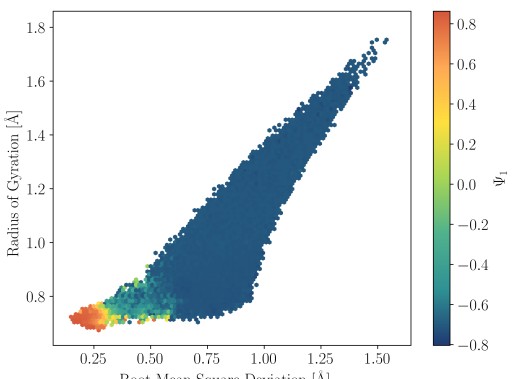
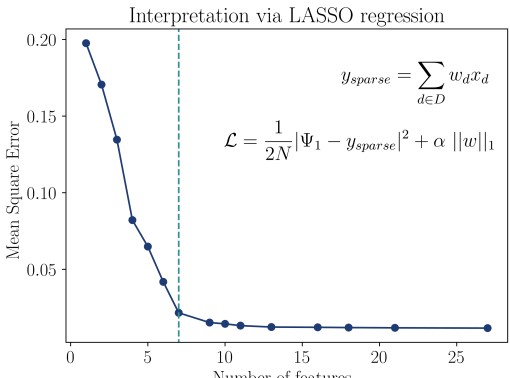

Figure 5: The value of the leading eigenfunction $\Psi_1$ of the evolution operator is highly correlated with the RMSD and Radius of Gyration of the Trp-cage protein.

Figure 6: MSE of approximating $\Psi_1$ by LASSO regression on meaningful physical descriptors. For Trp-cage we constructed a library of hydrogen-bond contact functions. The selected descriptors are reported in Fig. 7

| Physical descriptors (H-bonds) | Normalized Coefficient |
|---|---|
| GLY10-O – SER13-N | 0.307 |
| GLY11-O – ARG16-N | 0.294 |
| TRP6-O – GLY11-N | 0.170 |
| TRP6-NE1s (sidechain) – ARG16-O | 0.109 |
| GLN5-O – ASP9-N | 0.073 |
| TRP6-NE1s (sidechain) – PRO17-O | 0.044 |
| TRP6-NE1s (sidechain) – PRO18-N | 0.002 |

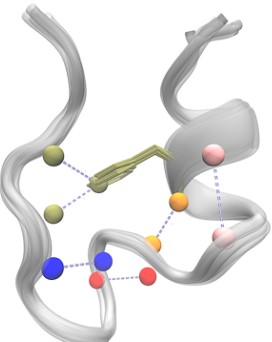

Figure 7: Normalized hydrogen-bond coefficients selected by the LASSO model (left) and representative structural snapshot (right) with the features highlighted.

In our simulations, we applied a funnel restraint (Limongelli et al., 2013) to limit the conformational space explored by the ligand in the unbound state, in turn accelerating the binding process. The parameters are identical to those used in previous studies (Rizzi et al., 2021). We define $h$ as the projection of each ligand along the binding axis, treated as its radial component. For $h \geq 10\,\text{Å}$, the funnel surface is a cylinder with radius $R_{\text{cyl}} = 2\,\text{Å}$ along the vertical axis. For $h < 10\,\text{Å}$, the funnel opens into a conical shape with a $45°$ angle, defined by $r = 12 - h$. The force acting on a displacement $x$ from the funnel surface is harmonic:

$$F_{\text{funnel}} = -k_F x \quad \text{with} \quad k_F = 20\,\text{kJ}\,\text{mol}^{-1}\,\text{Å}^{-2}$$

An additional harmonic restraint prevents the ligand from escaping too far from the host, enforcing an upper boundary:

$$F_{\text{upper}} = -k_U(h - 18) \quad \text{for} \quad h > 18\,\text{Å}, \quad \text{with} \quad k_U = 40\,\text{kJ}\,\text{mol}^{-1}\,\text{Å}^{-2}$$

The data will be released to ensure the reproducibility of the experiment.

**Training details.** Our encoder consisted of a SchNet (Schütt et al., 2017) graph neural network with 3 interaction blocks, 16 RBF functions, and a hidden dimension of 64 with an AdamW optimizer with starting learning rate of $10^{-2}$ decaying to $10^{-4}$ with a cosine schedule.

**Additional analysis.** In Fig. 8 we inspect the two leading eigenfunctions of the evolution operator by correlating them with two physical descriptors connected to the binding: the distance along the

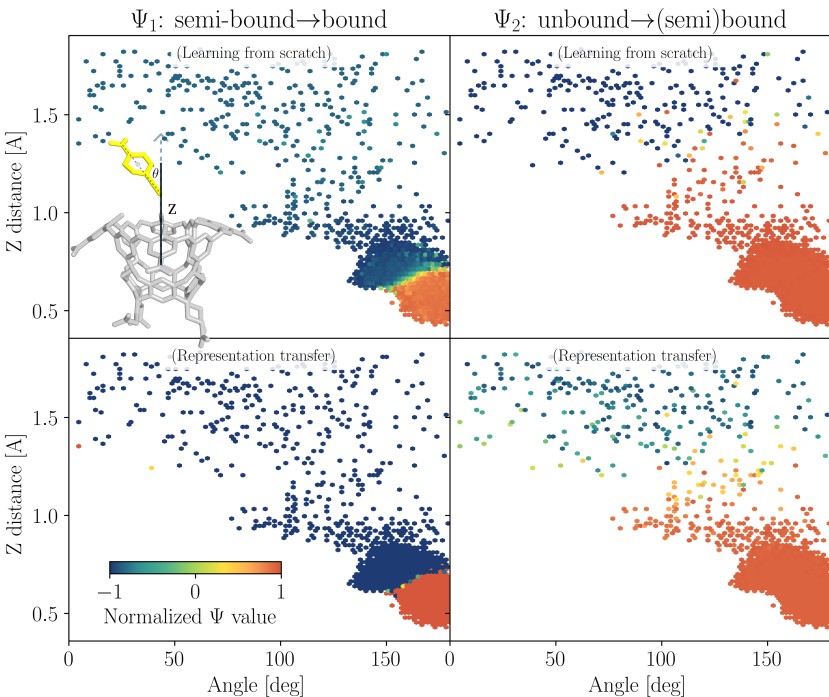

Figure 8: Analysis of the leading eigenfunctions in the space of the host-guest distance $z$ and the ligand orientation $\theta$ for $\Psi_1$ (left) and $\Psi_2$ (right). The first row contains the results obtained from training from scratch the representation, while the second row contains the case in which it is transferred from other systems.

$z$ direction between the center of mass of the host and the guest and the angle of the ligand with respect to the $z$ axis (see figure in the inset). These results allow us to correlate the $\Psi_1$ eigenfunction to the transition between the semi-bound pose to the native one, which is due to the presence of trapped water molecules inside the pocket (Rizzi et al., 2021; Bhakat & Söderhjelm, 2017). The second eigenfunction $\Psi_2$ is instead associated with the binding process. Furthermore, we compared the eigenfunctions obtained by training the representation from scratch on the G2 ligand with the case in which this is transferred from other ligands (G1 and G3), obtaining a remarkable agreement. The ligands G1, G2, and G3 are represented in Fig. 9

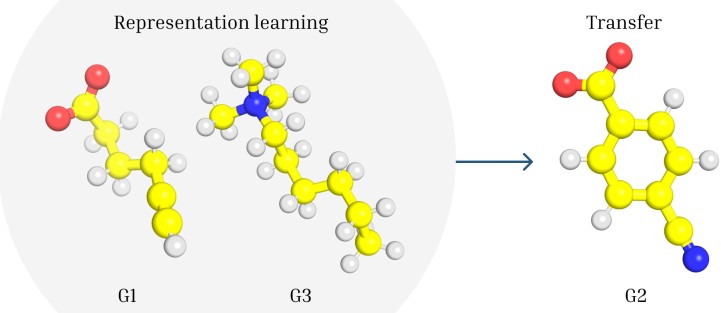

Figure 9: The three different molecules studied in the ligand-binding experiment.

## B.4 CLIMATE MODELING

**Datasets.** Following the methodology outlined in (NCP Center), we compute SST* from sea surface temperature (SST) data provided by the ORAS5 reanalysis (Zuo et al., 2019), as made available through the ChaosBench dataset (Nathaniel et al., 2024). The dataset spans a 45-year period

(1979–2023) at a spatial resolution of 1.5°, resulting in a time series of 540 monthly snapshots, each with dimensions $121 \times 240$. Data from 1979 to 2016 was used for training, while the 2017–2023 period was reserved for validation.

For the transfer learning task, we employed simulations from the CESM Last Millennium Ensemble project (Otto-Bliesner et al., 2016), spanning the years 850–2006 (files `b.e11.BLMTRC5CN.f19_g16.001.pop.h.SST.085001-089912.nc` to `b.e11.BLMTRC5CN.f19_g16.001.pop.h.SST.185001-200512.nc` available here: `https://gdex.ucar.edu/datasets/d651058/#`). To ensure spatial compatibility between synthetic and observational data, the CESM SST fields were regridded onto the same $1.5° \times 1.5°$ regular latitude–longitude grid of ORAS5 using the `xESMF` (Zhuang et al., 2025) python package.

**Training details.** Both models trained with CESM and ORAS5 data use a lightweight CNN encoder with four convolutional layers, batch normalization, and max pooling. A masked global average pooling layer, leveraging a binary land–ocean mask, ensures only ocean data contribute to the output representation. The pooled features are mapped through a final linear embedding layer.

For the CESM model, the linear layer $P$ maps to a 128-dimensional latent space. Training included simplicial normalization (Lavoie et al., 2022) (dimension 2), spectral normalization (Miyato et al., 2018) on the linear layer $P$, gradient clipping (max norm 0.2), a lag time of one month, 100 epochs, AdamW optimizer, and a cosine-decayed learning rate from $10^{-3}$ to $10^{-5}$ with a batch size of 64. Leading eigenvalues of the transfer operator are reported in Tab. 2.

Table 2: Leading eigenvalues of the transfer operator learned on ORAS5 data with $\varphi$ trained on CESM data. Each eigenvalue is expressed in terms of its real (Re), imaginary (Im), and absolute (Abs) components. The associated decorrelation times and oscillation frequencies (in years) are also reported. Eigenvalues are listed in descending order with respect to their absolute value, and those with a decorrelation time shorter than 1/12 years, i.e., the sampling frequency, were discarded.

| Idx | Re | Im | Abs | Decor (yr) | Freq (yr) | Idx | Re | Im | Abs | Decor (yr) | Freq (yr) |
|---|---|---|---|---|---|---|---|---|---|---|---|
| 6 | 0.92 | 0.00 | 0.92 | 1.01 | 0.00 | 13 | 0.60 | 0.11 | 0.61 | 0.17 | 3.01 |
| 4 | 0.88 | 0.09 | 0.89 | 0.70 | 5.23 | 15 | 0.58 | 0.13 | 0.59 | 0.16 | 2.37 |
| 5 | 0.88 | -0.09 | 0.89 | 0.70 | -5.23 | 16 | 0.58 | -0.13 | 0.59 | 0.16 | -2.37 |
| 2 | 0.76 | 0.40 | 0.86 | 0.54 | 1.09 | 17 | 0.58 | 0.03 | 0.58 | 0.15 | 9.78 |
| 3 | 0.76 | -0.40 | 0.86 | 0.54 | -1.09 | 18 | 0.58 | -0.03 | 0.58 | 0.15 | -9.78 |
| 7 | 0.85 | 0.00 | 0.85 | 0.50 | 0.00 | 19 | 0.52 | 0.12 | 0.53 | 0.13 | 2.34 |
| 8 | 0.79 | 0.00 | 0.79 | 0.35 | 0.00 | 20 | 0.52 | -0.12 | 0.53 | 0.13 | -2.34 |
| 0 | 0.41 | 0.67 | 0.78 | 0.34 | 0.52 | 21 | 0.47 | 0.15 | 0.49 | 0.12 | 1.68 |
| 1 | 0.41 | -0.67 | 0.78 | 0.34 | -0.52 | 22 | 0.47 | -0.15 | 0.49 | 0.12 | -1.68 |
| 9 | 0.74 | 0.12 | 0.75 | 0.29 | 3.26 | 23 | 0.47 | 0.03 | 0.47 | 0.11 | 9.27 |
| 10 | 0.74 | -0.12 | 0.75 | 0.29 | -3.26 | 24 | 0.47 | -0.03 | 0.47 | 0.11 | -9.27 |
| 11 | 0.71 | 0.00 | 0.71 | 0.24 | 0.00 | 31 | 0.39 | 0.00 | 0.39 | 0.09 | 0.00 |
| 12 | 0.64 | 0.00 | 0.64 | 0.19 | 0.00 | 27 | 0.35 | 0.16 | 0.38 | 0.09 | 1.19 |
| 14 | 0.60 | -0.11 | 0.61 | 0.17 | -3.01 | 28 | 0.35 | -0.16 | 0.38 | 0.09 | -1.19 |

For the ORAS5 model, the linear layer $P$ maps to a 256-dimensional latent space. Training details were otherwise identical, except a 12-month input history was used.

The hyperparameters reported above were selected via grid search; Tab. 3 summarizes the ranges explored.

**Comparisons.** We further compared our method to VAMPNets (Mardt et al., 2018), DPNets (Kostic et al., 2024b), Linear Least Squares (LinLS), and Kernel Ridge Regression (KRR) with a Gaussian kernel. For the deep-learning methods, we used identical training parameters across models. For the classical approaches applied to the raw inputs, we selected the best model via a grid search over regularization strengths $\alpha \in [10^{-7}, 10^{-3}]$ and, for KRR, kernel coefficients $\gamma \in [10^{-5}, 10^{-2}]$. We also varied the estimator rank in the set $\{5, 8, 10, 16, 32, 50, 64, 128\}$ to assess if low-rank approximations in the raw space could recover the dynamics. As shown in Tab. 6 and Fig. 13, our method outperforms both baselines on the evaluated tasks.

Table 3: Hyperparameter ranges explored during grid search for the climate modeling task.

| Hyperparameter | Search Range |
|---|---|
| Latent dimensions | `[32, 64, 128, ..., 1024]` |
| Max gradient clipping norm | `[None, 0.1, 0.2, 0.5]` |
| Normalization of linear layer | `[False, True]` |
| Regularization | `[0, 1e-5, ..., 1e-2]` |
| Simplicial normalization dimensions | `[0, 2, ..., 16]` |
| History length | `[0, 1, 2, 3, 6, 12]` |

### B.5 ABLATIONS

In our first set of ablations, we investigated the dependence of our self-supervised scheme on the encoder's architecture. Specifically, we studied the scaling of the loss function with respect to (i) the latent dimension and (ii) the overall parameter count of the encoder.

**Scaling laws: Graph-NN encoder.** We retrained the SchNet architecture Schütt et al. (2017) on the data from the protein folding experiment Sec. 4.1 for three different sizes of the encoder, summarized in Tab. 4, and values of the latent dimension from 4 to 256. The results of this comprehensive ablation study are reported in Fig. 10. We observed monotonically improving losses with respect to both an increasing number of training dimensions (panel A) and an increasing model size (panel B). This result provides robust confirmation of the good scalability properties of the loss function (8) studied in this work. As a test-time metric, we evaluated the eigenvalue residuals, as defined in (Colbrook et al., 2023, Algorithm 1), see panel C of Fig. 10. This metric assesses the extent to which the eigenvalues obtained from our model satisfy the eigenvalue equation $\mathsf{E}g = \lambda g$. The leading eigenvalue $\lambda_1$ is the one enjoying the overall best approximation. Larger architectures are associated with smaller residuals across all the leading eigenvalues.

Table 4: Architectural configuration of the three SchNet model sizes used in the ablation study.

| Model | Layers | Filters | Hidden Channels | Params |
|---|---|---|---|---|
| SchNet-S | 2 | 16 | 32 | 6,480 |
| SchNet-M | 3 | 32 | 64 | 33,088 |
| SchNet-L | 3 | 64 | 128 | 125,536 |

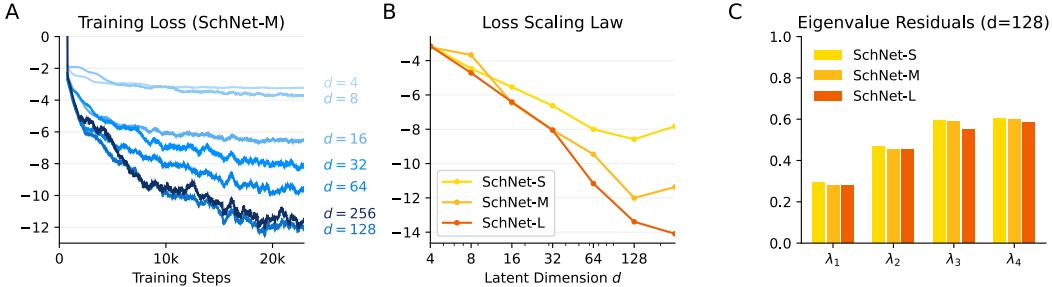

Figure 10: Scaling laws for the protein folding experiment in 4.1. **A** Training loss dynamics as a function of the number of latent dimensions $d$. **B** Final training loss for three different model sizes, as a function of the number of latent dimensions. **C** Eigenvalue residuals (lower is better), defined in (Colbrook et al., 2023, Algorithm 1) for three different model sizes.

**Scaling laws: CNN encoder.** The same set of ablations for the climate experiment Sec. 4.3 with a convolutional NN encoder, are reported in Fig. 11. The overall qualitative behavior exactly matches what was already observed for the Graph NN encoder: increasing latent dimensions and/or model size

(see Tab. 5) is associated with higher performance. To rule out the possibility that these improvements are linked to overfitting, in Fig. 11, we report the validation loss, instead of the training loss of Fig. 10.

Obtaining the same qualitative results across such distinct physical domains provides strong empirical evidence for the generality of the self-supervised method we propose.

Table 5: Architectural configuration of the three CNN model sizes used in the ablation study.

| Model | Layers | Hidden Channels | Params |
|-------|--------|-----------------|--------|
| CNN-S | 4 | [8, 16, 24, 32] | 12,888 |
| CNN-M | 4 | [16, 32, 64, 128] | 101,760 |
| CNN-L | 4 | [32, 64, 128, 256] | 397,024 |

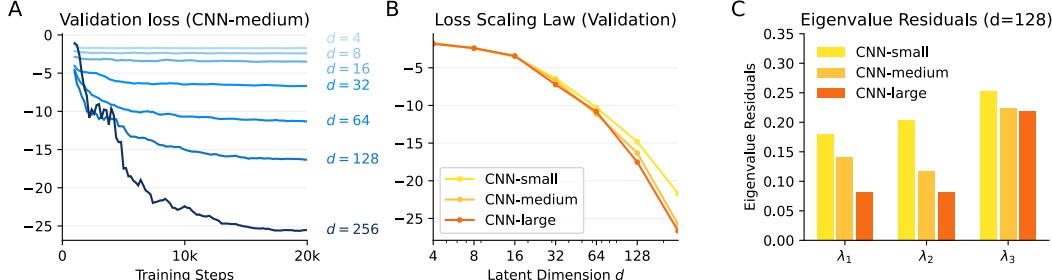

Figure 11: Scaling laws for the climate experiment in 4.3. **A** Validation loss dynamics as a function of the number of latent dimensions $d$. **B** Final validation loss for three different model sizes, as a function of the number of latent dimensions. **C** Eigenvalue residuals (lower is better), defined in (Colbrook et al., 2023, Algorithm 1) for three different model sizes.

**Online versus offline covariances.** We conducted an ablation study to assess the effect of using covariances $C_X, C_{XY}$ computed either online during training via EMA or offline from the full training set when estimating the evolution operator $E_\varphi$.

In the Lorenz '63 experiment, we trained the models as in the main Lorenz-63 experiment (see Appendix B.1) except for lag time set to 1 to enable a direct comparison between covariance estimation methods. The results show that online covariances yielded better performance, with RMSE and MAE of $0.51 \pm 0.11$ and $0.30 \pm 0.06$, respectively, compared to $0.63 \pm 0.21$ and $0.45 \pm 0.19$ for offline covariances.

In the climate experiment, the ENSO mode is easily recovered with both approaches. Specifically, for the model trained on ORAS5, the Pearson correlation between the right eigenfunction of $E_\varphi$ and the ONI was 0.72 with online covariances and 0.71 with offline covariances, indicating comparable performance. The associated eigenvalues were also very similar: $\lambda_{\text{ENSO}} = 0.9531 \pm 0.1206i$ (online) and $\lambda_{\text{ENSO}} = 0.9527 \pm 0.1277i$ (offline).

**Stability of EMA covariance.** To assess how EMA-based covariances converge toward their offline counterparts, computed via a full-pass over the entire training set, we measured their discrepancy in terms of Frobenius norm, i.e., $\|C_{\text{EMA}} - C_{\text{full-pass}}\|_F$, during training on the Lorenz '63 data. As shown in Fig. 12, this difference peaks in correspondence with the step-like drop in the validation loss, which we interpret as the encoder discovering new representational directions. For a sufficiently large number of epochs, as the network converges and settles into a stable representation, the discrepancy steadily decreases and approaches zero. These observations demonstrate how EMA offers a robust and practical online approximation of the offline, full-pass covariance, offering a clear advantage when dealing with large-scale datasets where computing full-pass covariances may be computationally infeasible.

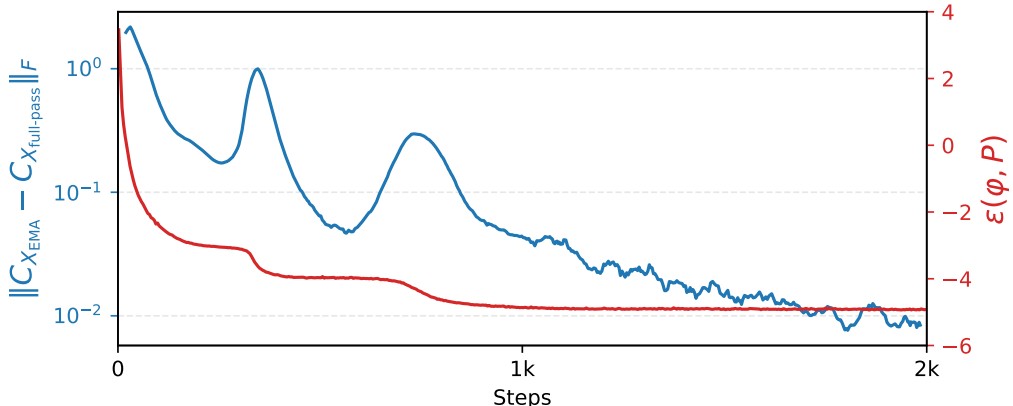

Figure 12: Stability of EMA-based covariance during training on Lorenz '63. The blue curve (left y-axis) shows the discrepancy between EMA and full-pass covariances, while the red curve (right y-axis) shows the validation loss.

Table 6: Performance comparison in terms of Pearson correlation between the right eigenfunction associated with the ENSO mode and ONI, alongside the time per training epoch. Best results are highlighted in bold.

**Transfer learning task (model trained on CESM, evaluated on ORAS5).**

|  | Ours | VAMPNets | DPNets | LinLS | KRR |
|---|---|---|---|---|---|
| Pearson correlation ($r$) | **0.81** | 0.56 | 0.77 | N/A | N/A |
| Time per epoch (s) | **25.27 ± 0.74** | 28.44 ± 0.79 | 29.17 ± 0.79 | N/A | N/A |

**Model trained and evaluated on ORAS5 data.**

|  | Ours | VAMPNets | DPNets | LinLS | KRR |
|---|---|---|---|---|---|
| Pearson correlation ($r$) | **0.72** | 0.56 | 0.62 | 0.60 | 0.63 |
| Time per epoch (s) | **1.91 ± 0.17** | 2.03 ± 0.15 | 2.05 ± 0.15 | N/A | N/A |

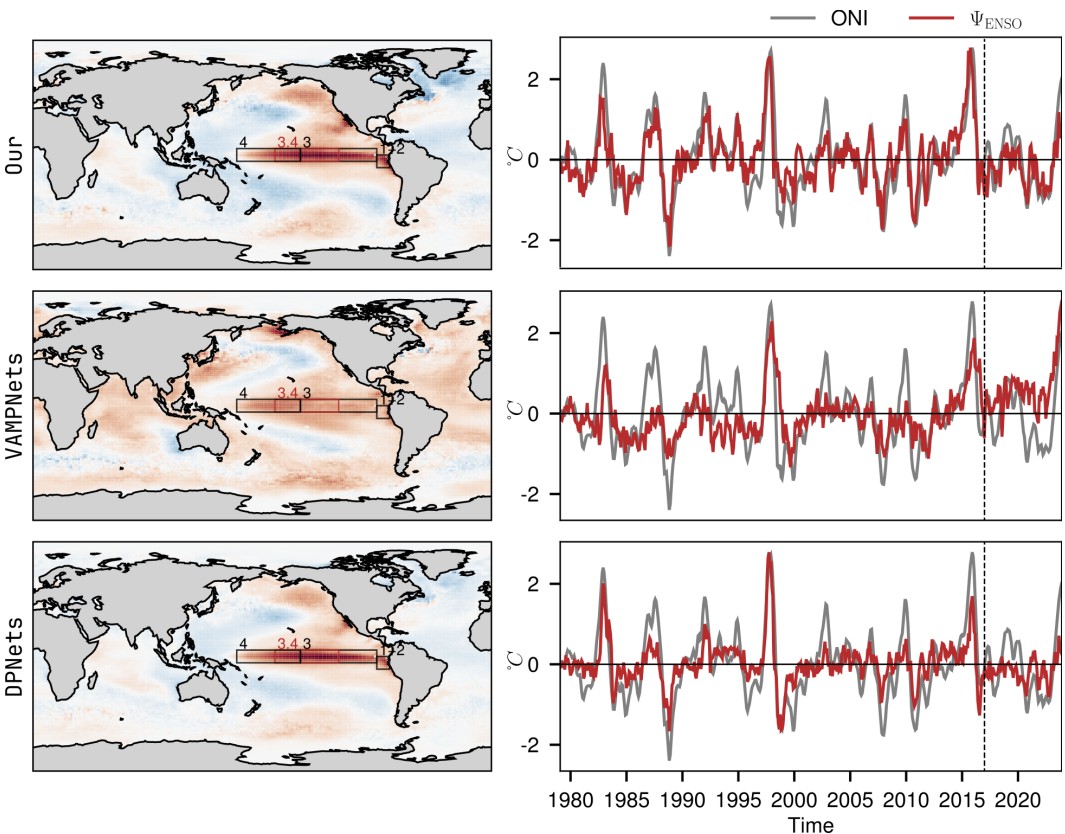

Figure 13: Comparison of ENSO modes retrieved using transfer learning by our method, VAMPNets, and DPNets.

