# OpenReview forum: "Self-Supervised Evolution Operator Learning for High-Dimensional Dynamical Systems"
_ICLR.cc/2026/Conference — ICLR 2026 Poster_

### Official Review · Reviewer_eZUJ · 2025-10-18

**Soundness:** 3
**Presentation:** 3
**Contribution:** 3
**Rating:** 8
**Confidence:** 3

**Summary:**

This paper presents an end-to-end deep learning framework for learning evolution operators of high-dimensional dynamical systems. It establishes a novel connection between self-supervised contrastive learning and operator learning theory.

Through experiments, the paper demonstrates that the proposed method can extract meaningful dynamical modes that reflect physically relevant processes such as folding-to-unfolding transition, molecular binding, and climate oscillation. It shows that these learned representations can be reused effectively across related systems.

**Strengths:**

The paper established a novel and rigorous theoretical connection between evolution operator learning and self-supervised contrastive learning. A major contribution is the scalability of the proposed framework to high-dimensional system. Empirically, the model demonstrates stable training and meaningful spectral decomposition in different scientific domians, even in terabyte-scale dataset.

**Weaknesses:**

(1) The finite-sample analysis is not provided;
(2) The paper lacks a clear comparison against other (Koopman) operator-learning approaches (e.g., kernel EDMD, Neural Operator frameworks).

**Questions:**

(1) Your Lemmas assume the evolution operator $E$ is Hilbert-Schmidt, yet you acknowledge this is often violated by deterministic dynamical systems. The Lorenz '63 system in your experiments is deterministic (or near-deterministic). So, what happens to the approximation quality when $E$ is not Hilbert-Schmidt?
(2) In the deterministic case, the Koopman operator acts on delta functions. How does your bilinear model $\langle \phi(x_t), P\phi(x_{t+1}) \rangle$ approximate this structure?
(3) How does the approximation error related to the latent dimension?
(4) In the climate experiment, how does your method perform if one applies a kernel-based approach directly to the raw 29,040-dimensional input features, instead of using the learned 128-dimensional embedding? Such a comparison would clarify if standard dimensionality reduction techniques could achieve similar results.
(5) Could you discuss any scenarios or system characteristics under which the learned representations may fail to transfer effectively?

---

> ### Author Response · Authors · 2025-11-18
> **Rebuttal:**
>
> **Weaknesses**
>
> > The finite-sample analysis is not provided;
>
> We thank the reviewer for raising this point. While a full finite-sample analysis is indeed beyond the scope of our work, primarily because jointly characterizing the approximation and estimation errors of neural networks remains a challenging open problem, we note that partial guarantees for the specific loss we use are available. In particular, Proposition 4.1 of _Spectral Representation Learning for Conditional Moment Models_ by Wang et al, and Theorem 2 of _Neural Conditional Probability for Uncertainty Quantification_ by Kostic et al. provide relevant finite-sample results.
> > The paper lacks a clear comparison against other (Koopman) operator-learning approaches (e.g., kernel EDMD, Neural Operator frameworks).
>
> We compare our approach against both linear least squares, that is classical DMD, and kernel ridge regression, that is kernel DMD, in the Lorenz ‘63 experiment reported in Appendix B.1. Additionally, following your third question, we have now included this comparison for the climate experiment as well. In both cases, our method demonstrates superior performance, highlighting how the learned representations capture non-linear features of the dynamics that are not readily accessible via classical operator-learning approaches applied directly to the raw data.
>
> ** Questions**
>
> > Your Lemmas assume the evolution operator is Hilbert-Schmidt, yet you acknowledge this is often violated by deterministic dynamical systems. The Lorenz '63 system in your experiments is deterministic (or near-deterministic). So, what happens to the approximation quality when is not Hilbert-Schmidt?
>
> Thank you for raising this point. Motivated by your comment, as well as related remarks from other reviewers, we conducted a more detailed analysis of how our method behaves when the evolution operator is not Hilbert–Schmidt. This situation indeed arises in many deterministic (or nearly deterministic) dynamical systems, including the Lorenz ’63 system discussed in Appendix B. Interestingly, we found that the self-supervised loss we adopt remains tied to approximating the evolution operator even when the underlying operator E is not Hilbert–Schmidt. We have added a proof and extended discussion of these observations in Section 3 of the main text and in Appendix A.1. In short: regardless of whether E is Hilbert–Schmidt, our self-supervised approach (i) approximates the evolution operator restricted to the space of functions spanned by the encoder, and (ii) encourages encoders $\varphi$ that, on average, capture the modes with the highest dynamical variability in the system. To the best of our knowledge, these observations are new, and we believe they significantly strengthen the theoretical foundation of our approach, especially in the non–Hilbert–Schmidt setting relevant for deterministic systems.
>
> > In the deterministic case, the Koopman operator acts on delta functions. How does your bilinear model approximate this structure?
>
> Thank you for this interesting question. Because our estimator is learned from finite, noisy data and uses regularization, it produces a broadened (smoothed) approximation of any delta-shaped conditional probability even when the true dynamics are deterministic. In the limit of an expressive feature map (e.g. random Fourier features with the number of features $\to \infty$, or infinite-width/NTK limits) and with appropriate regularization and sample growth, the bilinear form $\langle\varphi(y), P\varphi(x)\rangle$ can approximate arbitrarily sharp conditional structure: practically, we approximate the density-ratio projected onto the feature space, and recover the point-mass transport only in the infinite-feature / infinite-sample limit.
>
> > How does the approximation error related to the latent dimension?
>
> The dependence is directly tied to the previous point. Our estimator represents the density ratio $p(y\mid x)/p(y)$ through the bilinear form $\langle \varphi(y), P\varphi(x)\rangle$. For many common encoder architectures like random Fourier features, wide ReLU networks, and NTK limits, the feature map $\varphi$ is universal in the sense that, as the latent dimension $d \to \infty$, the span of the $\varphi_{i}$s becomes dense in the corresponding RKHS/function space. Consequently, the approximation error
> $
> \inf_{P} \| r - \langle \varphi(\cdot), P\varphi(\cdot)\rangle \|
> $
> tends to zero as $d\to\infty$, provided the target density ratio $r(x,y)$ has finite complexity in that space (e.g., finite RKHS norm). With finite $d$, this error is simply the projection error of $r$ onto the chosen feature space.

---

> > ### Author Response · Authors · 2025-11-18
> > **Rebuttal: last 2 questions.**
> >
> > > In the climate experiment, how does your method perform if one applies a kernel-based approach directly to the raw 29,040-dimensional input features, instead of using the learned 128-dimensional embedding? Such a comparison would clarify if standard dimensionality reduction techniques could achieve similar results.
> >
> > We thank the reviewer for suggesting this baseline. It is indeed crucial to verify if the performance gain stems from our learned embedding or simply from the application of operator-theoretic tools. Per your suggestion, we applied both DMD and kernel DMD directly to the raw 29,040-dimensional input features. We conducted an extensive hyperparameter search covering regularization strength $\alpha$, kernel bandwidth $\gamma$ (for RBF), and estimator rank constraints. As detailed in the revised Table 6 of the appendix B.4, our method ($r=0.72$) outperforms both linear least squares (that is classical DMD) ($r=0.60$) and Kernel ridge regression (that is KernelDMD) ($r=0.63$). This confirms that the learned 128-dimensional embedding captures non-linear features of the dynamics that are not easily accessible via standard kernel methods on the raw high-dimensional state.
> >
> > > Could you discuss any scenarios or system characteristics under which the learned representations may fail to transfer effectively?
> >
> > From a mathematical standpoint, transfer will fail when the features relevant to the new, unseen task are orthogonal (or nearly orthogonal) to the features captured by the self-supervised loss. In our case, the loss is designed to encode the slow modes of the training system. Therefore, if two physical systems exhibit slow modes governed by fundamentally different mechanisms, the learned representation may not generalize. For example, many condensed-matter systems are dominated by short-range atomic interactions and van der Waals forces, whereas protein conformational dynamics crucially depend on long-range electronic effects. This mismatch suggests that transferring representations between such classes of systems could be ineffective.

---

> > > ### Comment · Reviewer_eZUJ · 2025-11-25
> > >
> > > Thank you for your reply. I have read both comments and they mostly answered my questions. I'd main my score and raise up the confidence.

---

### Official Review · Reviewer_3ogW · 2025-10-28

**Soundness:** 2
**Presentation:** 2
**Contribution:** 2
**Rating:** 4
**Confidence:** 3

**Summary:**

This paper proposes an end-to-end, self-supervised approach to learning evolution operators for high-dimensional nonlinear dynamical systems, with a particular focus on scientific domains such as molecular dynamics and climate modeling. The authors connect contrastive, self-supervised representation learning techniques with the operator-theoretic framework, providing theoretical grounds for their approach and demonstrating its equivalence to established least-squares operator learning methods. Comprehensive experiments are reported on protein folding, molecular binding, and climate data, with claims of scalability, interpretability via spectral decomposition, and transferability of learned representations.

**Strengths:**

- The authors make a deliberate connection between self-supervised contrastive learning and classical operator-theoretic approaches (notably, the least-squares estimation of evolution operators). This is not only highlighted at an intuitive level but also underpinned with theoretical results.
- The paper supports claims with diverse, high-dimensional benchmarks, spanning molecular simulations (protein folding, ligand binding) and challenging climate datasets.
- The spectral decomposition of the learned operator yields interpretable, physically relevant modes, e.g., hydrogen bonding patterns in protein folding.

**Weaknesses:**

- While the core mathematical exposition is sound, there are places where notation could be clarified for easier accessibility:
  - In Lemma 2 and Appendix A, the formula for the predictor $P_*$ involves both $C_X^{-1}$ and $C_Y^{-1}$, but the mapping from the loss gradient to the closed-form $P_*$ is sketched rather than fully elucidated.
  - Equation (9) relates the action of E to the covariance of futures, but implementation choices for non-stationary or non-ergodic systems are not fully discussed.
- The limitations section (Conclusion) briefly notes the Hilbert-Schmidt assumption and qualitative evaluation. However, broader issues are not meaningfully discussed. For example, the treatment of deterministic versus stochastic systems, failure cases when the operator is not Hilbert-Schmidt, and scalability bottlenecks in training or memory for extremely large state spaces are not addressed.
- While the online/offline covariance ablation adds value, ablations for the choice of network architectures (GNN vs. CNN vs. MLP), size/sparsity of $P$, or impact of history/context window on model performance would strengthen the empirical narrative.

**Questions:**

- Can the authors provide (or reference) quantitative metrics for spectral decomposition accuracy on real-world, high-dimensional datasets, or propose benchmarks for such evaluation? For example, can clustering purity or precision/recall be reported for folded/unfolded distinctions in protein folding, or event detection in climate data?
- Can the authors report on representation transfer performance when the target system is more dissimilar to the source (e.g., transfer across molecular families, not just ligands, or different climate regimes)? What are the limits of transferability in practice?
- In extremely large-scale systems, does online or EMA-based covariance computation pose bottlenecks or stability issues? Can sparse or approximate methods be used safely?

---

> ### Author Response · Authors · 2025-11-18
> **Rebuttal: weaknesses**
>
> **Weaknesses**
>
> > While the core mathematical exposition is sound, there are places where notation could be clarified for easier accessibility: In Lemma 2 and Appendix A, the formula for the predictor involves both and , but the mapping from the loss gradient to the closed-form is sketched rather than fully elucidated.
>
> Thank you for bringing this issue to our attention. We have expanded the proof of Lemma 2 in Appendix A, showing the full derivation of the result. Furthermore, we have slightly clarified the statement of the Lemma, indicating the closed-form expression for the optimal predictor.
>
> > Equation (9) relates the action of E to the covariance of futures, but implementation choices for non-stationary or non-ergodic systems are not fully discussed.
>
> Our method does not require ergodicity nor stationarity; we highlighted this point explicitly at the beginning of Appendix A of the updated manuscript.
>
> > The limitations section (Conclusion) briefly notes the Hilbert-Schmidt assumption and qualitative evaluation. However, broader issues are not meaningfully discussed. For example, the treatment of deterministic versus stochastic systems, failure cases when the operator is not Hilbert-Schmidt, and scalability bottlenecks in training or memory for extremely large state spaces are not addressed.
>
> Thank you for this comment. Following your suggestion, and similar remarks from other reviewers, we expanded our discussion of the limitations concerning deterministic vs. stochastic systems and the Hilbert–Schmidt assumption. In particular, we analyzed more carefully what happens when the evolution operator is not Hilbert–Schmidt, which is common in deterministic settings. Remarkably, the self-supervised loss can be linked to approximating the evolution operator, even in this case. We added a proof and a concise discussion of this point and Appendix A.1 and Section 3.1, respectively..In short: even when E is not Hilbert–Schmidt, our self-supervised objective (i) approximates the evolution operator on the function space spanned by the encoder, and (ii) encourages encoders $\varphi$ that capture the modes with the highest dynamical variability.
>
> About possible scalability bottlenecks: as we stated below Lemma 3, our loss can be expressed entirely in terms of matrix multiplications, making it a perfect fit for GPU and TPU computing architectures commonly employed in ML and AI. This is in contrast to other prominent representation learning losses such as the VAMP loss [Mardt et al, 2018], which requires unwieldy matrix inversions, making the method both more expensive and numerically unstable. While we already had baselines showing the advantage of our methods on the VAMP score (see experiments 4.1, and 4.3), does the reviewer believe an expanded discussion on this point would be beneficial to the paper?
>
> > While the online/offline covariance ablation adds value, ablations for the choice of network architectures (GNN vs. CNN vs. MLP), size/sparsity of, or impact of history/context window on model performance would strengthen the empirical narrative.
>
> Thank you for raising these points. In the revised version of the manuscript, we have substantially expanded our ablation studies to address these concerns (see Appendix B.5). Specifically, we now include ablations over encoder architecture and embedding dimensionality, evaluating multiple architectures within the settings where they are appropriate (GNNs for protein-folding experiment and CNNs for climate data). These experiments show that smooth performance improvements as both model size and latent dimensionality increase, with diminishing returns for larger models and latent dimensions.
>
> [Mardt et al. 2018] _VAMPnets for deep learning of molecular kinetics_

---

> > ### Author Response · Authors · 2025-11-18
> > **Rebuttal: questions**
> >
> > **Questions**
> >
> > > Can the authors provide (or reference) quantitative metrics for spectral decomposition accuracy on real-world, high-dimensional datasets, or propose benchmarks for such evaluation? For example, can clustering purity or precision/recall be reported for folded/unfolded distinctions in protein folding, or event detection in climate data?
> >
> > Thank you for raising this important point. To the best of our knowledge, there is currently no established benchmark for evaluating the spectral decomposition accuracy of high-dimensional dynamical systems. For the climate experiment, we adopt Pearson correlation between ground-truth ONI index and the learned eigenfunction associated with the ENSO mode as a practical and interpretable proxy for evaluating the quality of the recovered spatial mode. In addition, in the new ablation studies included in the rebuttal, we report residuals as defined in [Colbrook et al. 2021], providing a quantitative metric for assessing the accuracy of the eigenvalue approximation (see Appendix B.5). Residuals therefore provide a principled metric showing how well the learned operator captures the leading spectral properties even in the absence of ground-truth eigenpairs.
> >
> > > Can the authors report on representation transfer performance when the target system is more dissimilar to the source (e.g., transfer across molecular families, not just ligands, or different climate regimes)? What are the limits of transferability in practice?
> >
> > Investigating how well the learned representations transfer to increasingly dissimilar dynamical systems is indeed an important next step. A thorough study of this question would require substantial work: training encoders on sufficiently broad and representative datasets, and defining evaluation metrics for the physical domain under investigation (drug design is clearly vastly different from climate modeling). In our view, this constitutes a separate project rather than an extension of the present manuscript. That said, we see our method as a foundational step to learn transferable representations for the physical sciences, and we expect the framework to be key to future studies on transferability limits.
> >
> > > In extremely large-scale systems, does online or EMA-based covariance computation pose bottlenecks or stability issues? Can sparse or approximate methods be used safely?
> >
> > We thank the reviewer for raising important points regarding the scalability and stability of EMA-based covariance computation in extremely large-scale systems. In the context of our work, covariance matrices are computed in the latent space, resulting in matrices of moderate size (latent_dimension × latent_dimension). Consequently, the computational and memory costs of maintaining these matrices do not pose significant bottlenecks in our experiments. Although larger batch sizes or very high latent dimensions could increase computational costs, such scenarios are still a negligible fraction of the full training step of the encoder network. Furthermore, for a sufficiently large number of epochs, once the network converges to a stable representation, EMA-based covariance estimates converge to their offline equivalents computed over the entire dataset. Additional evidence supporting this behavior is provided in Appendix B.5 (Ablations, Stability of EMA covariance). Given these considerations, we do not believe that sparse or approximate covariance estimation methods are necessary in our setting. Regarding scalability, we foresee that in multi-GPU training, it will be necessary to synchronize the estimated covariances across GPUs. This synchronization can be efficiently performed at the end of every training epoch, rather than at every training step, to minimize the overhead introduced by multi-GPU (and potentially multi-node) communication.

---

> ### Comment · Reviewer_3ogW · 2025-11-21
>
> I appreciate the authors providing the theoretical proof regarding the non-Hilbert-Schmidt case, which effectively addresses my concerns about the theoretical applicability to deterministic systems.
>
> As my primary concerns regarding soundness and evaluation have been meaningfully addressed, I am raising my score.

---

> > ### Author Response · Authors · 2025-11-24
> >
> > We thank you for your thoughtful feedback and for raising your score in light of our rebuttal. We are glad that the additional theoretical clarifications addressed your concerns and gave us the chance to show the applicability of our method even when the evolution operator is not Hilbert–Schmidt, which is common in deterministic systems.

---

### Official Review · Reviewer_FVgy · 2025-11-01

**Soundness:** 3
**Presentation:** 3
**Contribution:** 3
**Rating:** 6
**Confidence:** 3

**Summary:**

This paper introduces an encoder-only method for learning evolution operators of nonlinear dynamical systems by connecting self-supervised learning with operator theory. The approach is demonstrated across multiple domains, including protein folding, molecular binding, and climate pattern discovery.

**Strengths:**

1. The theoretical analysis provides a thorough and rigorous discussion of the paper's core claims.

2. The experimental design is strengthened by the use of multi-disciplinary datasets, ensuring the scenarios are both diverse and representative.

3. The integration of self-supervised representation learning with evolution operator theory represents a novel and promising research direction.

**Weaknesses:**

1. When the target function *f* does not lie in the linear span of the encoder, it is unclear how the method should be adjusted. The paper lacks discussion on this point—for instance, whether simply increasing the embedding dimension would suffice to satisfy this assumption.

2. The paper lacks practical guidance on how to choose the embedding dimension in real-world systems.

3. The authors should more clearly distinguish their approach from existing methods that combine deep learning with Koopman theory or DMD, and better highlight what additional problems their method can solve.

**Questions:**

1. How does your method handle cases where the target function *f* is not in the encoder's linear span?

2. What is the key advantage of your method over existing Koopman or DMD-based approaches, and what new problem does it enable you to solve?

---

> ### Author Response · Authors · 2025-11-18
> **Rebuttal**
>
> **Weaknesses**
> >  When the target function f does not lie in the linear span of the encoder, it is unclear how the method should be adjusted. The paper lacks discussion on this point—for instance, whether simply increasing the embedding dimension would suffice to satisfy this assumption.
>
> Thank you for raising this important point. The least squares estimator we obtain from the trained encoder can indeed act only on the linear span of the encoder.  When f is not contained in this space, one has two options:
>
> 1. Explicitly include $f$ as a non-trainable component of the encoder, as done in our experiment on the Lorenz ‘63 system (see Appendix B1).
> 2. If evaluations of $f$ on the training dataset are available (as usually the case), compute the least-squares estimator of $\hat{f}$ on the span of $\varphi$, and apply the learned operator to that. We note that this option is routinely used by the Kernel literature on evolution operators, see for example  [Kostic et al. 2021, Section 4].
>
> As you suggested, we added a discussion on this point in Section 3.1 of the revised version of the manuscript.
>
> > The paper lacks practical guidance on how to choose the embedding dimension in real-world systems.
>
> We thank the reviewer for bringing this point to our attention. In the revised manuscript, we now include a detailed set of ablations (see Appendix B.5, Figs. 10-11) that provide practical guidance on selecting the embedding dimension. Figures 10-11A/B, in particular, show that the loss function consistently converges as the embedding dimension increases, indicating that overshooting the optimal dimension does not harm performance. Do the reviewer believe it would be beneficial to practitioners to add an explicit discussion on this point?
>
> > The authors should more clearly distinguish their approach from existing methods that combine deep learning with Koopman theory or DMD, and better highlight what additional problems their method can solve.
>
> Thank you for the helpful comment. Our approach differs from existing deep-learning Koopman-based and DMD methods in several key aspects. In contrast to prior approaches that rely on both an encoder and a decoder, our method is encoder-only, which reduces the number of parameters and simplifies training. Moreover, the self-supervised contrastive loss we use is numerically stable and fully compatible with multi-GPU training, which allows the method to scale efficiently to high-dimensional dynamical systems. Additionally, the learned representations can also be readily transferred to related dynamical systems, enabling forms of transfer learning that classical DMD methods do not support. Finally, our framework naturally accommodates structural priors, such as graph-based or convolutional encoders, which cannot be easily incorporated into classical linear operator-learning techniques like DMD or kernel DMD. We have added a sentence at the end of Section 2 to emphasize these distinctions.
>
> **Questions**
>
> > How does your method handle cases where the target function f is not in the encoder's linear span?
>
> See our discussion on the first weakness, above.
>
> > What is the key advantage of your method over existing Koopman or DMD-based approaches, and what new problem does it enable you to solve?
>
> The key advantage of our method is that links the theoretical guarantees of DMD-based (that is, least-squares-based)  estimators with the expressive power of deep, self-supervised representations. Classical Koopman/DMD methods are well-understood in a statistical learning sense, and their accuracy is known to be critically linked to the representation . This happens because DMD methods fit a linear model directly in the latent space, which should be expressive enough to capture most of the dynamical variability of the process. Deep-learning approaches like [Lusch et al. 2018] can construct rich latent spaces, but the learned maps are not constrained to be valid evolution operators and thus carry no spectral or stability guarantees.
>
> Our method provides a formal mechanism that bridges both worlds by tying self-supervised representation learning to operator learning: the representation is learned for the purpose of preserving the linearity of the underlying evolution, and the operator is estimated with a provably consistent DMD-style procedure in that learned space. This enables, for the first time, learning evolution operators on large-scale scientific data (such as atomic graphs or multispectral fields) while retaining the interpretability and theoretical guarantees of classical Koopman/DMD approaches.
>
> [Kostic et al. 2021] _Learning dynamical systems via Koopman operator regression in reproducing kernel Hilbert spaces_.
>
> [Lusch et al. 2018] _Deep learning for universal linear embeddings of nonlinear dynamics_.

---

> > ### Comment · Reviewer_FVgy · 2025-11-27
> >
> > Thank the author for the detailed rebuttal, which has addressed all my concerns.

---

### Official Review · Reviewer_NFuM · 2025-11-05

**Soundness:** 2
**Presentation:** 3
**Contribution:** 2
**Rating:** 4
**Confidence:** 3

**Summary:**

This paper presents a self-supervised approach to learn the evolution operators, which characterize the temporal dynamics of complex stochastic/deterministic systems. Different from the forecasting or reconstruction type models, the method uses a contrastive self-supervised loss from the spectral contrastive learning to estimate the transfer operators and spectral decomposition directly from data.  The main contribution is to theoretically show the equivalence between the self-supervised loss and minimizing the operator regression objective, connecting the least-square estimator, VAMP-2 score, and HS norms. Experiments on three domains, including protein folding, molecular binding, and climate dynamics, show the effectiveness and generalization of the proposed method.

**Strengths:**

1. Theoretically explain the operator learning theory and self-supervised contrastive learning, which provides justification using contrastive objectives in scientific dynamic systems.
2. The method avoids computationally expensive matrix inversions and is implemented with simple matrix multiplications and covariance updates, which are very suitable for large-scale high-dimensional data.
3. The paper demonstrates the effectiveness through diverse and convincing experimental validation.

**Weaknesses:**

1. Although the qualitative results, such as the eigenfunction visualizations, look good. The paper does not report any standardized quantitative metrics and comparisons with the baseline on some experiments.
2. This paper is missing an ablation study that compares different encoder architectures or embedding dimensionalities.
3. Although admitted by authors that the Hilbert-Schmidt assumption is often not held in deterministic systems. The paper lacks a discussion of how the performance will degrade when the assumption fails. ( Can be one of your ablation studies.)

**Questions:**

1. It would be a benefit to elaborate more on Equation 8. How can this be viewed as a contrastive learning paradigm? Should state more if positive and negative pairs are not explicitly defined.

---

> ### Author Response · Authors · 2025-11-18
> **Rebuttal**
>
> **Weaknesses**
> > Although the qualitative results, such as the eigenfunction visualizations, look good. The paper does not report any standardized quantitative metrics and comparisons with the baseline on some experiments.
>
> We thank the reviewer for the comment. In addition to the qualitative visualizations of modes and eigenfunctions, the paper already reports quantitative metrics across all experiments. Specifically, we report:
>     - Implied timescales for protein folding experiment (see Section 4.1);
>     - Pearson correlation between the ground-truth ONI index and the learned eigenfunction associated with the ENSO mode for climate experiment (see Section 4.3 and Appendix B.4);
>     - Root-mean-square error (RMSE) and mean-absolute error (MAE) for Lorenz ‘63 experiment (see Appendix B.1).
>
> All of the above metrics are compared against multiple baselines. Additionally, in the new ablation studies conducted during the rebuttal, we also report eigenvalue residuals from [Colbrook et al. 2021] as a metric to assess the goodness of the eigenvalue approximation by our approach.
>
> > This paper is missing an ablation study that compares different encoder architectures or embedding dimensionalities. Thanks for this comment.
>
> In the revised version of the manuscript (Appendix B.5), we now report an extensive ablation study comparing the performance of our method across multiple embedding dimensions and encoder architectures. These ablations were run for both the GNN and CNN encoders in our protein-folding and climate experiments, respectively. The additional results, which we believe strengthen our method, show monotonically increasing performance as both the latent dimension and model size are increased, confirming our claims about the scalability properties of our approach. Moreover, by evaluating the eigenvalue approximation error via residuals, we confirm that the loss is highly effective at learning the leading eigenspace of the evolution operator, achieving accurate approximations of the leading eigenvalues even at low embedding dimensions.
>
> > Although admitted by authors that the Hilbert-Schmidt assumption is often not held in deterministic systems. The paper lacks a discussion of how the performance will degrade when the assumption fails. ( Can be one of your ablation studies.)
>
> Thank you for this comment. Prompted by it, as well as by related comments by the other reviewers, we investigated more in depth the failure modes of our method on non Hilbert-Schmidt operators. Remarkably, we have found out that the self-supervised loss we adopt remains linked to evolution operator learning even when the underlying operator E is not HS. We have included a proof and an expanded comment on these observations in Section 3.1 of the main text, and in Appendix A.1. The TLDR of our findings is that, irrespectively of the operator E being HS or not, our self-supervised approach on the one hand approximates the evolution operator restricted to the space of functions spanned by the encoder, while on the other hand promotes encoders $\varphi$ which, on average, capture the modes with highest dynamical variability in the system. In light of these observations, which, to the best of our knowledge, are novel, we believe that our approach is now significantly stronger.
>
> **Questions:**
> > It would be a benefit to elaborate more on Equation 8. How can this be viewed as a contrastive learning paradigm? Should state more if positive and negative pairs are not explicitly defined.
>
> Thank you for this question. In the first version of the manuscript, we briefly commented on the nature of positive pairs in Section 3.2 “Practical implementation”. That being said, we agree that the contrastive nature of Equation 8 is not self-evident, and in the updated version, we provided a comment on this point, right below its definition (Section 3).
>
> [Colbrook et al. 2021]  _Residual Dynamic Mode Decomposition: Robust and verified Koopmanism_

---

> > ### Comment · Reviewer_NFuM · 2025-11-25
> >
> > Thank you for your reply. The rebuttals answered my questions. I'd increase my score.

---

### Author Response · Authors · 2025-11-18
**Thank you for your review work, and updated manuscript.**

Dear Reviewers, we thank you for examining our work and for the many insights and feedback you have suggested in your reviews. While you can find a detailed answer for every point you raised in the dedicated response we wrote to each of you, we wish to highlight two main improvements of the revised paper resulting from your reviews:

1. **Substantial ablation studies on the dependence of our method on the latent dimension and architecture size.** You can find every relevant detail in the new Appendix B.5. A one-sentence summary of these new experiments is that the quality of the learned representations improves as a function of the expressivity of the encoder architecture. As both the model size and latent dimension are increased, one obtains lower losses and better eigenvalue approximations. This process hits diminishing returns as the model complexity grows, justifying the use of encoders and latent dimensions of moderate size.

2. **On the theoretical side, we have justified our method even in the case of non-Hilbert-Schmidt evolution operators, which commonly appear in deterministic dynamical systems.** As discussed in the revised Section 3.1, and the newly added Appendix A.1, our self-supervised loss function can always be linked to approximating the evolution operator, and in the non-Hilbert-Schmidt case it also promotes encoders capturing most of the dynamical variability of the process; see Appendix A.1 for a detailed derivation.

Thanks to these additions, inspired by your comments, we believe the paper is now substantially stronger, supported by rigorous theory, while being experimentally sound and robust. We have uploaded a revised version of the manuscript with the changes highlighted in red.

---

### Author Response · Authors · 2025-12-01
**Overview of Our Paper and Rebuttal Status**

Dear AC, reviewers, and readers,

Due to the reassignment of ICLR ACs following OpenReview’s security incident on November 27th, we are writing to provide a high-level overview of our paper and the rebuttal status.

Our work falls within the broad field of machine learning for dynamical systems and introduces an end-to-end framework for learning evolution operators (a.k.a. Koopman operators) of large-scale, nonlinear dynamical systems. Evolution operators are a mathematically sound paradigm for modeling nonlinear dynamical systems, whereby the system’s state is mapped through a representation that *linearizes the dynamics*. In this work, we bridged self-supervised representation learning with recent advancements in operator-learning theory, showing the equivalence between the *spectral contrastive loss* and evolution operator learning. This led us to develop a scalable and principled approach for learning evolution operators. Through an extensive set of experiments, covering protein dynamics, molecular binding, and climate modeling, we demonstrated how models trained with our approach can extract interpretable spatio-temporal structures from complex datasets. Every point raised by the reviewers in the initial round of reviews has been addressed during the rebuttal. In particular, as detailed in our general response to reviewers, our responses primarily focused on two aspects:

1. Ablation studies assessing the dependence of our method on latent space dimensionality and architecture size, showing that the quality of the learned representations improves as a function of the encoder architecture's expressivity.
2. A theoretical justification demonstrating that our self-supervised loss continues to approximate the evolution operator even when it is non-Hilbert-Schmidt, promoting encoders that capture the dominant dynamical variability of the process.

We want to highlight that reviewers reacted positively to our additional experiments and clarifications; in particular:

* Reviewer **NFuM** increased their score from 4 to 6 (confidence unchanged at 3);
* Reviewer **FVgy** maintained a score of 6 and confidence of 3;
* Reviewer **3ogW** increased their score from 4 to 6 (confidence unchanged at 3);
* Reviewer **eZUJ** maintained their score of 8 and increased their confidence from 3 to 5.

Finally, we emphasize that we posted our rebuttal on November 19, and the reviewers’ follow-up responses were submitted on November 21 (3ogW), November 25 (eZUJ, NFuM), and November 27 (FVgy).

With sincere gratitude for overseeing our submission,

The Authors

---

### Meta-Review · Area_Chair_NkEg · 2025-12-20

**Summary:**

The paper introduces a framework for learning evolution operators of high-dimensional dynamical systems by establishing a theoretical equivalence between spectral contrastive learning and operator regression. The proposed architecture is encoder-only and avoids the numerical instability of matrix inversions found in prior methods, enabling scaling effectively to complex scientific domains such as protein folding, molecular binding and climate modeling.

The reviewers agreed that the bridge between self-supervised representation learning with operator theory is interesting and its empirical success is sufficiently demonstrated on large-scale scientific datasets. The recommendation is for acceptance.

**Reviewer Concerns:**

Initial reviews identified weaknesses regarding a lack of ablation studies for encoder architectures, quantitative metrics and theoretical gaps regarding the handling of non-Hilbert-Schmidt operators in deterministic systems. The authors provided a comprehensive rebuttal that included the requested ablations confirming the method's robustness to latent dimensionality and model size. A new proof is added to justify the applicability to non-Hilbert-Schmidt operators. There are no obvious outstanding concerns.

**Reviewer Scores:**

All reviewers would have raised or maintained their scores above the accepting threshold based on their existing interactions.

---

### Decision · Program_Chairs · 2026-01-26

Accept (Poster)